# Analyzing Deep PAC-Bayesian Learning with Neural Tangent Kernel: Convergence, Analytic Generalization Bound, and Efficient Hyperparameter Selection

**Wei Huang**[*]  *wei.huang.vr@riken.jp*
*RIKEN Center for Advanced Intelligence Project (AIP)*

**Chunrui Liu**[*]  *chunrui.liu@student.uts.edu.au*
*University of Technology Sydney*

**Yilan Chen**  *yilan@ucsd.edu*
*University of California San Diego*

**Richard Yi Da Xu**  *xuyida@hkbu.edu.hk*
*Hong Kong Baptist University*

**Miao Zhang**  *miaozhang1991@gmail.com*
*Harbin Institute of Technology (Shenzhen)*

**Tsui-Wei Weng**  *lweng@ucsd.edu*
*University of California San Diego*

**Reviewed on OpenReview:** *https://openreview.net/forum?id=nEX2q5B2RQ*

## Abstract

PAC-Bayes is a well-established framework for analyzing generalization performance in machine learning models. This framework provides a bound on the expected population error by considering the sum of training error and the divergence between posterior and prior distributions. In addition to being a successful generalization bound analysis tool, the PAC-Bayesian bound can also be incorporated into an objective function for training probabilistic neural networks, which we refer to simply as *Deep PAC-Bayesian Learning*. Deep PAC-Bayesian learning has been shown to achieve competitive expected test set error and provide a tight generalization bound in practice at the same time through gradient descent training. Despite its empirical success, theoretical analysis of deep PAC-Bayesian learning for neural networks is rarely explored. To this end, this paper proposes a theoretical convergence and generalization analysis for Deep PAC-Bayesian learning. For a deep and wide probabilistic neural network, our analysis shows that PAC-Bayesian learning corresponds to solving a kernel ridge regression when the probabilistic neural tangent kernel (PNTK) is used as the kernel. We utilize this outcome in conjunction with the PAC-Bayes $\mathcal{C}$-bound, enabling us to derive an analytical and guaranteed PAC-Bayesian generalization bound for the first time. Finally, drawing insight from our theoretical results, we propose a proxy measure for efficient hyperparameter selection, which is proven to be time-saving on various benchmarks. Our work not only provides a better understanding of the theoretical underpinnings of Deep PAC-Bayesian learning, but also offers practical tools for improving the training and generalization performance of these models.

---

[*]Equal Contribution.

# 1 Introduction

Deep learning has demonstrated powerful learning capability due to its over-parameterization structure, in which various network architectures have been responsible for its significant leap in performance (LeCun et al., 2015). However, deep neural networks often suffer from over-fitting and complex hyperparameters, making the design of generalization guarantees a crucial research goal (Zhang et al., 2021). One recent breakthrough in this area is the development of a learning framework that connects geometry of the loss landscape with the flatness of minima such as Sharpness-Aware Minimization Foret et al. (2021) and Deep PAC-Bayesian learning, which trains a probabilistic neural network using an objective function based on PAC-Bayesian bounds (Bégin et al., 2016; Dziugaite & Roy, 2017; Neyshabur et al., 2018; Raginsky et al., 2017; Neyshabur et al., 2017; London, 2017; Smith & Le, 2018; Pérez-Ortiz et al., 2021; Guan & Lu, 2022). Deep PAC-Bayesian learning provides a tight generalization bound while achieving competitive expected test set error (Ding et al., 2022). Furthermore, this generalization bound can be computed from the training data, obviating the need to split data into training, testing, and validation sets, which is highly applicable when working with limited data (Pérez-Ortiz et al., 2021; Grünwald & Mehta, 2020).

The success of PAC-Bayesian learning has led to their widespread application in different deep neural network architectures, including convolutional neural networks (Zhou et al., 2019; Pérez-Ortiz et al., 2021), binary activated multilayer networks (Letarte et al., 2019), partially aggregated neural networks (Biggs & Guedj, 2021), and graph neural networks (Liao et al., 2021; Ju et al., 2023); different learning frameworks such as meta-learning (Amit & Meir, 2018; Rothfuss et al., 2021; Flynn et al., 2022), adversarial training and robustness Wang et al. (2022b), contrastive unsupervised learning (Nozawa et al., 2020). These advancements have enabled significant improvements in the generalization performance of deep neural networks, making PAC-Bayesian learning a valuable tool for training robust and reliable models.

The remarkable empirical success of PAC-Bayesian learning has generated growing interest in understanding its theoretical properties. However, theoretical investigations have been largely limited to specific variants of the technique, such as Entropy-SGD, which minimizes an objective indirectly by approximating stochastic gradient ascent on the so-called local entropy (Dziugaite & Roy, 2018a), and differential privacy (Dziugaite & Roy, 2018b). Other investigations have relied heavily on empirical exploration (Neyshabur et al., 2017). To the best of our knowledge, there has been no systematic investigation into why PAC-Bayesian learning is successful and why the PAC-Bayesian bound is tight on unseen data after training. For example, it is still unclear when applying gradient descent to PAC-Bayesian learning:

**Q1:** How effective is gradient descent training on a training set?

**Q2:** How tight is the generalization bound after convergence on the unseen dataset?

These questions are particularly important because understanding the strengths and weaknesses of PAC-Bayesian learning can help improve its theoretical foundations and practical applications. For instance, answering the first question can shed light on the training dynamics and optimization challenges of PAC-Bayesian learning, while answering the second question can provide insights into the generalization performance of this learning framework.

Answering the questions posed earlier can be challenging due to the inherent non-convexity of optimization in neural networks (Jain et al., 2017), the additional randomness introduced by probabilistic neural networks (Specht, 1990), and the challenges brought by the Kullback-Leibler divergence between posterior/prior distribution pairs. However, recent advances in deep learning theory with over-parameterized settings offer a promising approach to tackling these challenges. It has been shown that wide networks optimized with gradient descent can achieve near-zero training error, and the neural tangent kernel (NTK) plays a critical role in the training process. The NTK remains unchanged during gradient descent training (Jacot et al., 2018), providing a guarantee for achieving a global minimum (Du et al., 2019; Allen-Zhu et al., 2019). In the PAC-Bayesian framework, the NTK is calculated based on the gradient of the distribution parameters of the weights, rather than the derivative of the weights themselves. We refer to this as the Probabilistic NTK (PNTK), which we use to develop a convergence analysis for characterizing the optimization process of PAC-Bayesian learning. The PAC-Bayes bounds employed in Deep PAC-Bayesian learning typically stem

from the PAC-Bayes-kl theorem, as indicated by (Seeger, 2002). These bounds study the expected risk, often referred to as Gibbs risk. Conversely, an alternative line of PAC-Bayes bounds, known as the $\mathcal{C}$-bound Lacasse et al. (2006), presents an upper bound for the risk associated with the Majority Vote classifier, the risk of expected output. Examining the risk of expected output within the PAC-Bayes bound presents the benefit of providing an explicit solution for optimization analysis in the infinite-width limit. Moreover, we formulate the generalization bound of PAC-Bayesian learning upon convergence for the first time.

We summarize our contributions as follows:

- We provide a detailed characterization of the gradient descent training process of the PAC-Bayesian objective function and show that the final solution corresponds to kernel ridge regression with its kernel being the PNTK.

- Building upon the optimization solution, we derive an analytical and guaranteed PAC-Bayesian bound for deep networks after training for the first time. In contrast to other PAC-Bayesian bounds that require the distribution of the posterior, our bound, based on the $\mathcal{C}$-bound under a Gaussian posterior, where only the mean parameter is being trained, is entirely independent of computing the distribution of the posterior.

- We design a training-free proxy based on our theoretical bound to select hyperparameters efficiently, which is effective and time-saving. Our training-free proxy can help alleviate the computational burden of hyperparameter selection, which is critical for practical applications of PAC-Bayesian learning.

- Our technique of analyzing optimization and generalization of probabilistic neural networks through over-parameterization has a wide range of applications, such as the Variational Auto-encoder (Kingma & Welling, 2013; Rezende et al., 2014) and deep Bayesian networks (MacKay, 1992; Neal, 2012). We believe that our technique can provide the basis for the analysis of over-parameterized probabilistic neural networks.

## 2 Related Work

**PAC-Bayes Bound** The Probably Approximately Correct (PAC) Bayes framework (McAllester, 1999a;b) is a useful tool for providing a test performance (generalization) guarantee by incorporating knowledge about the learning algorithm and probability distribution over a set of hypotheses. This framework has been extended to the analysis of generalization bounds for probabilistic neural networks using the PAC-Bayesian method (Langford & Caruana, 2002). While the original PAC-Bayes theory only works with bounded loss functions, Germain et al. (2016) proposed a PAC-Bayes bound for sub-gamma loss family and Haddouche et al. (2021) expanded the PAC-Bayesian theory to learning problems with unbounded loss functions. Furthermore, several improved PAC-Bayesian bounds suitable for different scenarios have been introduced by Bégin et al. (2014; 2016). The flexibility and generalization properties of the PAC-Bayes framework make it a popular tool for analyzing complex, non-convex, and overparameterized optimization problems, particularly for over-parameterized neural networks (Guedj, 2019). In the context of feedforward neural networks with ReLU activations, Neyshabur et al. (2018) presented a generalization bound based on the product of the spectral norm of the layers and the Frobenius norm of the weights. This bound has been used as a benchmark for evaluating the generalization performance of various neural network architectures and training methods.

**PAC-Bayesian Learning.** Obtaining a numerical bound on generalization is just as important as achieving a theoretical analysis for the generalization properties of deep learning. One method for computing an error upper bound is by training a Bayesian neural network and using a refined PAC-Bayesian bound, as introduced by Langford & Caruana (2002). Building on this work, Neyshabur et al. (2017) developed a training objective function derived from a relaxed PAC-Bayesian bound to train deep neural networks. In standard PAC-Bayes, the prior is often chosen to be a spherical Gaussian centered at the origin. However, this approach might not be effective in cases where the KL divergence is unreasonably large due to a lack of

information about the data. To address this issue, a growing body of literature proposes to obtain localized PAC-Bayes bounds via distribution-dependent priors informed by the data (Ambroladze et al., 2007; Negrea et al., 2019; Dziugaite et al., 2021; Perez-Ortiz et al., 2021). Additionally, Dziugaite & Roy (2018b); Tinsi & Dalalyan (2022) showed how a differentially private data-dependent prior can yield a valid PAC-Bayes bound in situations where the data distribution is unknown. Recently, researchers have been focused on providing PAC-Bayesian bounds for more complex and realistic neural network architectures, such as convolutional neural networks (Zhou et al., 2019), binary activated multilayer networks (Letarte et al., 2019), partially aggregated neural networks (Biggs & Guedj, 2021), attention-based neural networks (Wang et al., 2022b), and graph neural networks (Liao et al., 2021; Ju et al., 2023). In this work, our goal is to demystify the success of deep PAC-Bayesian Learning by exploring the role of the Probabilistic Neural Tangent Kernel (PNTK) in achieving a tight generalization bound.

## 3 Preliminary

**Notation.** In this paper, we use bold-faced letters for vectors and matrices (e.g., $\mathbf{x}$ and $\mathbf{W}$), and non-bold-faced letters for scalars (e.g., $n$ and $t$). We use $\| \cdot \|_2$ to denote the Euclidean norm of a vector and the spectral norm of a matrix, while $\| \cdot \|_F$ denotes the Frobenius norm of a matrix. For a neural network, we use $\sigma(x)$ to denote the activation function, which applies an element-wise nonlinearity to its input. We denote the set $1, 2, \ldots, n$ as $[n]$, and we use $\lambda_0(\mathbf{A}) = \lambda_{\min}(\mathbf{A})$ to denote the least eigenvalue of matrix $\mathbf{A}$. Throughout the paper, we will also use other common notation, such as $O(\cdot)$ for asymptotic notation and $\mathbb{E}[\cdot]$ for the expected value of a random variable. We will define any additional notation as needed.

### 3.1 Deep probabilistic neural network

In PAC-Bayesian learning, we use probabilistic neural networks (PNNs) instead of deterministic networks. In PNNs, the weights follow a certain distribution, and in this work, we use the Gaussian distribution. We define an $L$-layer PNN $f$ governed by the following recursive expression:

$$\mathbf{x}^{(l)} = \frac{1}{\sqrt{m}} \sigma\big(\mathbf{W}^{(l)} \mathbf{x}^{(l-1)}\big), \quad 1 \leq l \leq L-1; \quad f = \frac{1}{\sqrt{m}} \mathbf{W}^{(L)} \mathbf{x}^{(L-1)}, \tag{1}$$

where $m$ is width of hidden layers, $\mathbf{x}^{(0)} = \mathbf{x} \in \mathbb{R}^d$ is the input with $d$ being the input dimension, $\mathbf{W}^{(1)} \in \mathbb{R}^{m \times d}$ is the weight matrix at the first layer, $\mathbf{W}^{(l)} \in \mathbb{R}^{m \times m}$ is the weight at the $l$-th layer for $2 \leq l \leq L-1$, and $\mathbf{W}^{(L)} \in \mathbb{R}^{m \times 1}$ is the weight at the output layer.

We employ the reparameterization trick (Kingma & Welling, 2013; Kingma et al., 2015) to model the weights as Gaussian during gradient descent training, with trainable mean and variance parameters:

$$\mathbf{W}^{(l)} = \mathbf{W}_\mu^{(l)} + \mathbf{W}_\sigma^{(l)} \odot \boldsymbol{\xi}^{(l)}, \ \boldsymbol{\xi}^{(l)} \sim \mathcal{N}(\mathbf{0}, \mathbf{I}), \ 1 \leq l \leq L, \tag{2}$$

where $\odot$ denotes the element-wide product operation. In this expression, for all $1 \leq l \leq L$, $\boldsymbol{\xi}^{(l)}$ share the same size as their corresponding weight matrix or vector. The re-parameterization trick involves sampling $\boldsymbol{\xi}^{(l)}$ for $1 \leq l \leq L$ from a normal distribution $\mathcal{N}(\mathbf{0}, \mathbf{I})$.

We randomly initialize the mean weights as follows: $\mathbf{W}_\mu^{(l)} \sim \mathcal{N}(\mathbf{0}, c_\mu^2 \cdot \mathbf{I})$ for $l \in [1, L]$. Here, we omit the size of the mean $\mathbf{0}$ and variance $\mathbf{I}$, which is determined by the corresponding weight matrix. For the variance weights, we use an absolute constant to initialize them, namely, $\mathbf{W}_\sigma^{(l)} = c_\sigma^2 \cdot \mathbf{1}$, where $\mathbf{1}$ is a matrix with all elements equal to 1. Our assumption of an isotropic initialization for the covariance matrix is based on the empirical setting presented in Rivasplata et al. (2019).

### 3.2 PAC-Bayes bound

Suppose we have a set of $n$ i.i.d. samples $\mathcal{S} = \{(\mathbf{x}_i, y_i)\}_{i=1}^n$ drawn from a non-degenerate distribution $\mathcal{D}$. Let $\mathcal{H}$ denote the hypothesis space and $h(\mathbf{x})$ the prediction of hypothesis $h \in \mathcal{H}$ for input $\mathbf{x}$. The generalization error of classifier $h$ with respect to $\mathcal{D}$ is denoted by $R_{\mathcal{D}}(h) = \mathbb{E}_{(\mathbf{x},y) \sim \mathcal{D}}[\ell(y, h(\mathbf{x}))]$, where $\ell(\cdot)$ is a given loss function. The empirical error of classifier $h$ with respect to $\mathcal{S}$ is denoted by $R_{\mathcal{S}}(h) = \frac{1}{n} \sum_{i=1}^n \ell(y_i, h(\mathbf{x}_i))$.

In PAC-Bayes, the prior distribution $Q(0) \in \mathcal{H}$ represents the distribution over the hypothesis space $\mathcal{H}$ before training or at initialization, and the posterior distribution $Q \in \mathcal{H}$ represents the distribution of parameters after training. The expected population risk and empirical error are defined as $R_{\mathcal{D}}(Q) = \mathbb{E}_{(\mathbf{x},y)\sim\mathcal{D}, h\sim Q}[\ell(y, h(\mathbf{x}))] = \mathbb{E}_{h\sim Q}[R_{\mathcal{D}}(h)]$ and $R_{\mathcal{S}}(Q) = \mathbb{E}_{h\sim Q}[R_{\mathcal{S}}(h)]$, respectively.

The PAC-Bayes theory (Seeger, 2002; Maurer, 2004) provides the following theorem:

**Theorem 3.1.** *For any $\delta \in (0, 1]$, any loss function $\ell : \mathbb{R} \times \mathbb{R} \to [0, 1]$, the following inequality holds uniformly for all posteriors distributions $Q \in \mathcal{H}$ with a probability of at least $1 - \delta$,*

$$\mathrm{kl}\big(R_{\mathcal{S}}(Q)\|R_{\mathcal{D}}(Q)\big) \leq \frac{\mathrm{KL}(Q\|Q(0)) + \log\frac{2\sqrt{n}}{\delta}}{n}, \tag{3}$$

*where $\mathrm{KL}(Q\|Q(0)) = \mathbb{E}_Q\left[\ln\frac{Q}{Q(0)}\right]$ is the Kullback-Leibler (KL) divergence and $\mathrm{kl}(q\|q') = q\log(\frac{q}{q'}) + (1 - q)\log(\frac{1-q}{1-q'})$ is the binary KL divergence.*

Furthermore, by combining Pinsker's inequality for binary KL divergence, $\mathrm{kl}(\hat{p}\|p) \geq (p - \hat{p})^2/(2p)$, where $\hat{p} < p$, we can obtain the following bound:

$$R_{\mathcal{D}}(Q) - R_{\mathcal{S}}(Q) \leq \sqrt{2R_{\mathcal{D}}(Q)\frac{\mathrm{KL}(Q\|Q(0)) + \log\frac{2\sqrt{n}}{\delta}}{n}}. \tag{4}$$

Equation (4) is a classical result. This result can be further combined with the inequality $\sqrt{ab} \leq \frac{1}{2}(\bar{\lambda}a + \frac{b}{\bar{\lambda}})$, for all $\bar{\lambda} > 0$, which leads to a PAC-Bayes-$\lambda$ bound in Theorem 3.2, as proposed by Thiemann et al. (2017):

**Theorem 3.2.** *Let $Q(0) \in \mathcal{H}$ be some prior distribution over $\mathcal{H}$. Then for any $\delta \in (0, 1]$, the following inequality holds uniformly for all posteriors distributions $Q \in \mathcal{H}$ with a probability of at least $1 - \delta$*

$$R_{\mathcal{D}}(Q) \leq \frac{R_{\mathcal{S}}(Q)}{1 - \bar{\lambda}/2} + \frac{\mathrm{KL}(Q\|Q(0)) + \log\frac{2\sqrt{n}}{\delta}}{n\bar{\lambda}(1 - \bar{\lambda}/2)}. \tag{5}$$

Another line of PAC-Bayes $\mathcal{C}$-bound research (Laviolette & Marchand, 2007; Laviolette et al., 2011; Germain et al., 2015) focuses on the upper bound of the $Q$-weighted majority vote classifier $B_Q = \mathrm{sng}(\mathbb{E}_{h\sim Q}[h(\mathbf{x}_i)])$, also known as the Bayes classifier. In this context, we define the $Q$-margin realized on an example $(\mathbf{x}, y)$, which is an important notion related to majority votes: $\mathcal{M}_Q(\mathbf{x}, y) = y \cdot \mathbb{E}_{h\sim Q}h(\mathbf{x})$. We also consider the first moment $\mathcal{M}_Q^{\mathcal{D}} = \mathbb{E}_{(\mathbf{x},y)\sim\mathcal{D}}\mathcal{M}_Q(\mathbf{x}, y)$ and the second moment $\mathcal{M}_{Q^2}^{\mathcal{D}} = \mathbb{E}_{(\mathbf{x},y)\sim\mathcal{D}}\mathbb{E}_{(h,h')\sim Q^2}h(\mathbf{x})h'(\mathbf{x})$ of the $Q$-margin. The generalization bound of $R_{\mathcal{D}}(B_Q)$ is given as follows:

**Theorem 3.3** (PAC Bayes $\mathcal{C}$-Bound (Laviolette et al., 2011)). *Let $Q \in \mathcal{H}$ be any distribution over $\mathcal{H}$. Then for any $\delta \in (0, 1]$, for any $n \geq 8$, for any auto-complemented family $\mathcal{H}$ of $B$-bounded real value functions, the following inequality holds with a probability of at least $1 - \delta$:*

$$R_{\mathcal{D}}(B_Q) \leq 1 - \frac{(\mathcal{M}_Q^{\mathcal{D}})^2}{\mathcal{M}_{Q^2}^{\mathcal{D}}} \leq 1 - \frac{(\mathcal{M}_Q^{\mathcal{S}} - 2B\frac{\sqrt{\ln(\frac{2n}{\delta})}}{\sqrt{2n}})^2}{\mathcal{M}_{Q^2}^{\mathcal{S}} + 2B^2\frac{\sqrt{\ln(\frac{2n}{\delta})}}{\sqrt{2n}}}, \tag{6}$$

*where $\mathcal{M}_Q^{\mathcal{S}} = \frac{1}{n}\sum_{i=1}^n \mathcal{M}_Q(\mathbf{x}_i, y_i)$ and $\mathcal{M}_{Q^2}^{\mathcal{S}} = \frac{1}{n}\sum_{i=1}^n \mathbb{E}_{(h,h')\sim Q^2}h(\mathbf{x}_i)h'(\mathbf{x}_i)$.*

The $\mathcal{C}$-bound is an upper bound of the risk of Majority vote classifier, that depends on the first two moments of the margin of the $Q$-convex combination realized on the data. While the PAC-Bayes bounds (Theorems 3.1 and 3.2) deal with classifiers that use a fixed subset of the training examples. In contrast, the $\mathcal{C}$-bound that applies to a stochastic average (and a majority vote) of classifiers using different subsets (of different sizes) of the training examples. In particular, the $\mathcal{C}$-bound is proposed for the restricted case where the voters are all classifiers, which is consistent with the kernel ridge regression derived in this work.

### 3.3 Deep PAC-Bayesian learning

In this work, we aim to use PAC-Bayes theory to design the training objective for PNNs, inspired by previous work such as Catoni (2007); Rivasplata et al. (2019). Specifically, we choose Equation (5) as the training objective, which is a PAC-Bayes bound that guarantees generalization error. It is worth noting that the original interest of Theorem 3.2 in (Thiemann et al., 2017) was to allow the optimization of a quasiconvex objective with respect to both $\bar{\lambda}$ and $Q$. However, since our main goal is to study the optimization and generalization properties of PNNs, we set $\bar{\lambda} = 1$ and omit the factor of two. Moreover, drawing inspiration from the Bayes classifier, which takes the expectation over $Q$ inside the loss function, we define the objective function as follows:

$$L(Q) = \underline{R}_{\mathcal{S}}(Q) + \lambda \frac{\mathrm{KL}(\boldsymbol{\theta}(t) \| \boldsymbol{\theta}(0))}{n} = \frac{1}{n} \sum_{i=1}^{n} \ell\left(y_i, \hat{f}\left(\mathbf{x}_i\right)\right) + \lambda \frac{\mathrm{KL}(\boldsymbol{\theta}(t) \| \boldsymbol{\theta}(0))}{n}, \tag{7}$$

where $\hat{f}(t) \triangleq \mathbb{E}_{f \sim Q}[f(\mathbf{x}; t)]$ is an expected output function, $\lambda$ is a hyperparameter introduced in a heuristic manner to make the method more flexible, and $\boldsymbol{\theta}$ is the collection of all weights. Contrasting the generalization bound presented by Dziugaite & Roy (2017), which computes the average over the training set using random samples, our approach utilizes the empirical loss of the expected output function inspired by the PAC-Bayes $\mathcal{C}$-bound. In particular, the empirical loss $\underline{R}_{\mathcal{S}}(Q) \triangleq \frac{1}{n} \sum_{i=1}^{n} \ell\left(y_i, \hat{f}\left(\mathbf{x}i\right)\right)$ used in Equation (7) is a lower bound of the expected empirical loss (also known as Gibbs empirical risk) $R_{\mathcal{S}}(Q)$, as per Jensen's inequality. This approach offers the advantage of yielding an explicit solution for the expected output function in the infinite-width limit. To train the PNN, we use the reparameterization trick to update the mean and variance of the weight matrices and vectors. Specifically, the update rule for $\mathbf{W}_{\mu}^{(l)}$ and $\mathbf{W}_{\sigma}^{(l)}$ with $1 \leq l \leq L$ at iteration $t$ is given by:

$$\mathbf{W}_{\mu}^{(l)}(t+1) = \mathbf{W}_{\mu}^{(l)}(t) - \eta \frac{\partial L(Q)}{\partial \mathbf{W}_{\mu}^{(l)}(t)}; \quad \mathbf{W}_{\sigma}^{(l)}(t+1) = \mathbf{W}_{\sigma}^{(l)}(t) - \eta \frac{\partial L(Q)}{\partial \mathbf{W}_{\sigma}^{(l)}(t)}, \tag{8}$$

where $\eta$ is the learning rate. In our theoretical analysis, we consider the gradient flow instead of gradient descent, but the same results can be extended to the gradient descent case with a careful analysis.

## 4 Main Theoretical Results

In this section, Theorem 4.2 gives a precise characterization of how the objective function without KL divergence decreases to zero. We then extend the convergence characterization to the full objective, and find the final solution is a kernel ridge regression, as demonstrated by Theorem 4.3. As a consequence, we are able to establish an analytic generalization bound through Theorem 4.4.

### 4.1 Optimization analysis

We begin by simplifying the analysis to focus on optimizing probabilistic neural networks of the form (1) with the objective function $\underline{R}_{\mathcal{S}}(Q)$, neglecting the KL divergence term for now. We show that the results obtained for this simplified setting can be extended to the target function with KL divergence in the next section. In particular, the objective function can be expressed as follows:

$$\underline{R}_{\mathcal{S}}(Q; t) = \frac{1}{2n} \|\mathbf{y} - \mathbb{E}_{f \sim Q} f(\mathbf{X}; t)\|_2^2 = \frac{1}{2n} \left\|\mathbf{y} - \hat{f}(\mathbf{X}; t)\right\|_2^2, \tag{9}$$

where $\mathbf{X} = \{\mathbf{x}_i\}_{i=1}^{n} \in \mathbb{R}^{n \times d}$ represents inputs and the corresponding label is $\mathbf{y} = \{y_i\}_{i=1}^{n} \in \mathbb{R}^n$. Given this premise, we start by establishing that, for a $L$-layer probabilistic neural network of the form (1), the gradient flow of expected output function admits the following dynamics:

$$\frac{d\hat{f}(\mathbf{X}; t)}{dt} = \frac{\partial \hat{f}(\mathbf{X}; t)}{\partial \boldsymbol{\theta}_{\mu}} \frac{\partial \boldsymbol{\theta}_{\mu}}{\partial t} + \frac{\partial \hat{f}(\mathbf{X}; t)}{\partial \boldsymbol{\theta}_{\sigma}} \frac{\partial \boldsymbol{\theta}_{\sigma}}{\partial t} = \frac{1}{n} \left(\mathbf{y} - \hat{f}(\mathbf{X}; t)\right) \left(\boldsymbol{\Theta}_{\mu}(\mathbf{X}, \mathbf{X}; t) + \boldsymbol{\Theta}_{\sigma}(\mathbf{X}, \mathbf{X}; t)\right), \tag{10}$$

where $\boldsymbol{\theta}_\mu \triangleq (\{\mathbf{W}_\mu^{(l)}\}_{l=1}^L)$, and $\boldsymbol{\theta}_\sigma \triangleq (\{\mathbf{W}_\sigma^{(l)}\}_{l=1}^L)$ are collection of mean weights and variance weights. Without loss of generality, we set the learning rate $\eta = 1$. Furthermore, $\boldsymbol{\Theta}_\mu(\mathbf{X}, \mathbf{X}; t) \in \mathbb{R}^{n \times n}$ and $\boldsymbol{\Theta}_\sigma(\mathbf{X}, \mathbf{X}; t) \in \mathbb{R}^{n \times n}$ are *probabilistic neural tangent kernels* (PNTKs), which are defined as follows:

**Definition 4.1** (Probabilistic Neural Tangent Kernel). *The tangent kernels associated with the expected output function $\hat{f}(\mathbf{X}; t)$ at parameters $\boldsymbol{\theta}_\mu$ and $\boldsymbol{\theta}_\sigma$ are defined as,*

$$
\begin{aligned}
\boldsymbol{\Theta}_\mu(\mathbf{X}, \mathbf{X}; t) &= \frac{\partial \hat{f}(\mathbf{X}; t)}{\partial \boldsymbol{\theta}_\mu} \left( \frac{\partial \hat{f}(\mathbf{X}; t)}{\partial \boldsymbol{\theta}_\mu} \right)^\top = \sum_{l=1}^L \nabla_{\mathbf{W}_\mu^{(l)}} \hat{f}(\mathbf{X}; t) \nabla_{\mathbf{W}_\mu^{(l)}} \hat{f}(\mathbf{X}; t)^\top, \\
\boldsymbol{\Theta}_\sigma(\mathbf{X}, \mathbf{X}; t) &= \frac{\partial \hat{f}(\mathbf{X}; t)}{\partial \boldsymbol{\theta}_\sigma} \left( \frac{\partial \hat{f}(\mathbf{X}; t)}{\partial \boldsymbol{\theta}_\sigma} \right)^\top = \sum_{l=1}^L \nabla_{\mathbf{W}_\sigma^{(l)}} \hat{f}(\mathbf{X}; t) \nabla_{\mathbf{W}_\sigma^{(l)}} \hat{f}(\mathbf{X}; t)^\top.
\end{aligned}
\tag{11}
$$

A few remarks on Definition 4.1 are in order. Unlike standard (deterministic) neural networks, a probabilistic network has two sets of parameters, namely, $\boldsymbol{\theta}_\mu$ and $\boldsymbol{\theta}_\sigma$, which lead to two corresponding tangent kernels. We introduce the expected output function $\hat{f}(t)$ in the definition to correspond to the expected empirical loss $R_{\mathcal{S}}(Q)$ (9), which helps us to address the additional randomization problem caused by PNNs. Explicitly the analytic forms of PNKT are given as follows:

$$
\boldsymbol{\Theta}_{\mu,ij}^{(l)}(0) = \sum_{r=1}^m \mathbb{E}_{\boldsymbol{\xi}} \left[ \frac{1}{\sqrt{m}} \frac{\partial \hat{f}(\mathbf{x}_i)}{\partial x_{i,r}^{(l)}} \sigma'((\mathbf{w}_r^{(l)})^\top \mathbf{x}_i^{(l-1)}) \right] \mathbb{E}_{\boldsymbol{\xi}} \left[ \frac{1}{\sqrt{m}} \frac{\partial \hat{f}(\mathbf{x}_j)}{\partial x_{j,r}^{(l)}} \sigma'((\mathbf{w}_r^{(l)})^\top \mathbf{x}_j^{(l-1)}) \right],
\tag{12}
$$

$$
\boldsymbol{\Theta}_{\sigma,ij}^{(l)}(0) = \sum_{r=1}^m \mathbb{E}_{\boldsymbol{\xi}} \left[ \frac{1}{\sqrt{m}} \frac{\partial \hat{f}(\mathbf{x}_i)}{\partial x_{i,r}^{(l)}} \sigma'((\mathbf{w}_r^{(l)})^\top \hat{\mathbf{x}}_i^{(l-1)}) \odot \boldsymbol{\xi}_r^{(l)} \right] \mathbb{E}_{\boldsymbol{\xi}} \left[ \frac{1}{\sqrt{m}} \frac{\partial \hat{f}(\mathbf{x}_i)}{\partial x_{j,r}^{(l)}} \sigma'((\mathbf{w}_r^{(l)})^\top \mathbf{x}_j^{(l-1)}) \odot \boldsymbol{\xi}_r^{(l)} \right].
\tag{13}
$$

One of the key contributions of this work is the observation that the PNTKs $\boldsymbol{\Theta}_\mu(\mathbf{X}, \mathbf{X})$ and $\boldsymbol{\Theta}_\sigma(\mathbf{X}, \mathbf{X})$ both converge to limiting *deterministic* kernels, at initialization and during training when $m$ is sufficiently large. Specifically, we have $\lim_{m \to \infty} \boldsymbol{\Theta}_\mu(\mathbf{X}, \mathbf{X}) = \boldsymbol{\Theta}_\mu^\infty(\mathbf{X}, \mathbf{X})$ and $\lim_{m \to \infty} \boldsymbol{\Theta}_\sigma(\mathbf{X}, \mathbf{X}) = \boldsymbol{\Theta}_\sigma^\infty(\mathbf{X}, \mathbf{X})$. This result is important because it implies that the training dynamics of PNNs can be analyzed in terms of the corresponding deterministic kernels.

Thanks to the convergence of the PNTKs to deterministic kernels as the network width goes to infinity, the gradient flow dynamics of the output function become linear in the infinite-width limit. Specifically, the dynamics take the form:

$$
\frac{d\hat{f}(\mathbf{X}; t)}{dt} = \frac{1}{n} \left( \mathbf{y} - \hat{f}(\mathbf{X}; t) \right) \left( \boldsymbol{\Theta}_\mu^\infty(\mathbf{X}, \mathbf{X}) + \boldsymbol{\Theta}_\sigma^\infty(\mathbf{X}, \mathbf{X}) \right).
\tag{14}
$$

This key finding allows us to establish our main convergence theory for deep probabilistic neural networks, which we state formally below.

**Theorem 4.2** (Convergence of probabilistic networks with large width). *Suppose $\sigma(\cdot)$ is $H$-Lipschitz, $\beta$-Smooth. Assume $\sigma(\cdot)$ and its partial derivative $\frac{\partial \sigma(x)}{\partial x}$ are continuous in $x$, $\lambda_0(\mathbf{K}_\infty^{(L)}) > 0$, and the network's width is of $m = \Omega\left( 2^{O(L)} \max\left\{ \frac{n^2 \log(Ln/\delta)}{\lambda_0^2(\mathbf{K}_\infty^{(L)})}, \frac{n}{\delta}, \frac{n^4}{\lambda_0^4(\mathbf{K}_\infty^{(L)})} \right\} \right)$. Then, with a probability of at least $1 - \delta$ over the random initialization, we have,*

$$
\underline{R}_{\mathcal{S}}(Q(t)) \le \exp\left( -\lambda_0(\mathbf{K}_\infty^{(L)})t \right) \underline{R}_{\mathcal{S}}(Q(0)),
\tag{15}
$$

*where we define NNGP kernel and its derivative as $K_{ij}^{(l)} \triangleq (\hat{\mathbf{x}}_i^{(l)})^\top \hat{\mathbf{x}}_j^{(l)}$ with $\hat{\mathbf{x}}^{(l)} = \mathbb{E}_{\boldsymbol{\xi}^{(l)}} \frac{1}{\sqrt{m}} \sigma\left( \mathbf{W}^{(l)} \hat{\mathbf{x}}^{(l-1)} \right)$, and in the infinite-width limit $\mathbf{K}_{ij,\infty}^{(l)} \triangleq \lim_{m \to \infty} \mathbf{K}_{ij}^{(l)}$ for $1 \le l \le L$.*

The proof sketch of Theorem 4.2 can be found in Section 5.1 and all the detailed proof are shown in Appendix A. Our main convergence theory for deep probabilistic neural networks is based on the insight that in the infinite-width limit, the dynamics of the output function with gradient flow is linear. Specifically, if $m$ is large

enough, our theorem establishes that the expected training error converges to zero at a linear rate, with the least eigenvalue of the NNGP kernel governing the convergence rate. Furthermore, we find that the change of weight is bounded during training, as shown in Lemma A.6, which is consistent with the requirement of the PAC-Bayes theory that the loss function is bounded.

## 4.2 Training with KL divergence

We expand the KL-divergence in the objective function (7) with $\boldsymbol{\theta}(t) \sim \mathcal{N}(\boldsymbol{\theta}_\mu(t), \boldsymbol{\theta}_\sigma(t))$:

$$\text{KL}(\boldsymbol{\theta}(t)\|\boldsymbol{\theta}(0)) = \frac{1}{2n} \sum_{i=1}^{P} \left( \log \frac{\theta_{\sigma,i}(0)}{\theta_{\sigma,i}(t)} + \frac{(\theta_{\mu,i}(t) - \theta_{\mu,i}(0))^2}{\theta_{\sigma,i}(0)^2} + \frac{\theta_{\sigma,i}(t)}{\theta_{\sigma,i}(0)} - 1 \right), \tag{16}$$

where $P$ is the number of parameters. Furthermore, we assume that the variance weights are fixed during training, which is empirically verified in Appendix C.3. With this assumption, the objective function (7) reduces to:

$$L(Q) = \frac{1}{2n} \left\| \mathbf{y} - \hat{f}(\mathbf{X}; t) \right\|_2^2 + \frac{\lambda}{2nc_\sigma^2} \|\boldsymbol{\theta}_\mu(t) - \boldsymbol{\theta}_\mu(0)\|_2^2, \tag{17}$$

where $\boldsymbol{\theta}_\mu(t) = (\{\mathbf{W}_\mu^{(l)}(t)\}_{l=1}^L, \mathbf{v}_\mu(t))$ is the collection of mean weights at time $t$, and $c_\sigma$ is a constant that controls the scale of variance weights. By analyzing the objective function (17), we arrive at the conclusion that in the infinite-width limit, a probabilistic neural network performs kernel ridge regression, as stated in the following theorem:

**Theorem 4.3.** *Suppose $m \geq \text{poly}(n, 1/\lambda_0, 1/\delta, 1/\mathcal{E})$ and the objective function follows the form (17), for any test input $\mathbf{x}_{te} \in \mathbb{R}^d$ with probability at least $1 - \delta$ over the random initialization, we have*

$$\hat{f}(\mathbf{x}_{te}; t)|_{t=\infty} = \boldsymbol{\Theta}_\mu^\infty(\mathbf{x}, \mathbf{X})(\boldsymbol{\Theta}_\mu^\infty(\mathbf{X}, \mathbf{X}) + \lambda/c_\sigma^2 \mathbf{I})^{-1} \mathbf{y} \pm \mathcal{E}, \tag{18}$$

*where $\mathcal{E}$ is a small error term between output function of finite-width PNN and infinite-width PNN.*

The detailed proof is provided in Section 5.2 and Appendix B. Theorem 4.3 sheds light on the regularization effect of the KL term in PAC-Bayesian learning and provides an explicit expression for the convergence of the output function.

## 4.3 Generalization analysis

We adopt a general and suitable loss function $\ell \in [0, 1]$ to evaluate the PNN's generalization, while using squared loss for training. Theorem 3.2 provides the PAC-Bayesian bound concerning the distribution at initialization and after optimization. By combining this result with Theorem 4.3, we can provide a generalization bound for PAC-Bayesian learning under ultra-wide conditions.

**Theorem 4.4** (PAC-Baye $\mathcal{C}$-bound with NTK). *Suppose that $\mathcal{S} = \{(\mathbf{x}_i, y_i)\}_{i=1}^n$ are i.i.d. samples from a non-degenerate distribution $\mathcal{D}$, and $m \geq \text{poly}(n, \lambda_0^{-1}, \delta^{-1})$. Then with a probability of at least $1 - \delta$ over the random initialization and the training samples, the probabilistic neural network (PNN) trained by gradient descent for $T \geq \Omega(\frac{1}{\eta \lambda_0} \log \frac{n}{\delta})$ iterations has $R_\mathcal{D}(B_Q)$ that is bounded as follows:*

$$R_\mathcal{D}(B_Q) \leq 1 - \frac{\left( \frac{1}{n} \sum_{i=1}^n y_i \hat{f}^\infty(\mathbf{x}_i) - 2B \frac{\sqrt{\ln(\frac{2n}{\delta})}}{\sqrt{2n}} \right)^2}{\frac{1}{n} \sum_{i=1}^n \sum_{j=1}^n \hat{f}^\infty(\mathbf{x}_i) \hat{f}^\infty(\mathbf{x}_j) + 2B^2 \frac{\sqrt{\ln(\frac{2n}{\delta})}}{\sqrt{2n}}}. \tag{19}$$

The proof can be found in Section B.1. Theorem 4.4 provides a generalization bound for the PAC-Bayesian learning framework under the ultra-wide condition, combining results from Theorem 3.3 and Theorem 4.3. Note that the generalization ability of kernel ridge regression has been studied in standard neural networks Hu et al. (2020); Nitanda & Suzuki (2021); Nonnenmacher et al. (2021). In this work, the bound is analytic and computable, in contrast to the PAC-Bayes bound (5), thus providing a theoretical guarantee for the PAC-Bayesian learning approach.

# 5 Proof Sketch

## 5.1 Proof of Theorem 4.2

To prove Theorem 4.2, we observe that if $\boldsymbol{\Theta}_\mu$ and $\boldsymbol{\Theta}_\sigma$ converge to a deterministic kernel, then the dynamics of output function admit a linear system, which is tractable during evolution. In this paper we focus on $\boldsymbol{\Theta}^{(L)} = \boldsymbol{\Theta}_\mu^{(L)} + \boldsymbol{\Theta}_\sigma^{(L)}$, the gram matrix induced by the weights from $L$-th layer for simplicity at the cost of a minor degradation in convergence rate. Before demonstrating the main steps, we introduce the expected neural network model, which is recursively expressed as follows:

$$\hat{\mathbf{x}}^{(l)} = \mathbb{E}_{\boldsymbol{\xi}^{(l)}} \frac{1}{\sqrt{m}} \sigma\big(\mathbf{W}^{(l)}\hat{\mathbf{x}}^{(l-1)}\big), \quad 1 \le l \le L-1; \quad \hat{f} = \mathbb{E}_{\boldsymbol{\xi}^{(L)}} \frac{1}{\sqrt{m}} \mathbf{W}^{(L)}\hat{\mathbf{x}}^{(L-1)}.$$

Note that this definition is in contrast to the definition as expected empirical loss (9) in PAC-Bayes. The Neural Network Gaussian Process (NNGP) for PNN (Lee et al., 2018) is defined as follows:

$$K_{ij}^{(l)} = (\hat{\mathbf{x}}_i^{(l)})^\top \hat{\mathbf{x}}_j^{(l)}, \quad K_{ij,\infty}^{(l)} = \lim_{m\to\infty} K_{ij}^{(l)},$$

where subscript $i, j$ denote the index of input samples. We aim to show that $\boldsymbol{\Theta}^{(L)}$ is close to $\mathbf{K}_\infty^{(L)}$ in large width. In particular, to prove Theorem 4.2, three core steps are:

Step 1 Show at initialization $\lambda_0\big(\boldsymbol{\Theta}^{(L)}(0)\big) \ge \frac{\lambda_0\big(\mathbf{K}_\infty^{(L)}\big)}{2}$ and the required condition on $m$.

Step 2 Show during training $\lambda_0\big(\boldsymbol{\Theta}^{(L)}(t)\big) \ge \frac{\lambda_0\big(\mathbf{K}_\infty^{(L)}\big)}{2}$ and the required condition on $m$.

Step 3 Show during training the expected empirical loss $R_\mathcal{S}(Q)$ has a linear convergence rate.

In our proof, we mainly focus on deriving the condition on $m$ by analyzing $\lambda_0\big(\boldsymbol{\Theta}^{(L)}(0)\big)$ at initialization through Lemma A.2 and Lemma A.3. For step 2, we construct Lemma A.4 Lemma A.5 and Lemma A.6 to demonstrate that $\lambda_0\big(\boldsymbol{\Theta}^{(L)}(t)\big) \ge \frac{\lambda_0(\mathbf{K}_\infty^{(L)})}{2}$. This leads to the required condition on $m$ during train. Finally, we summarize all the previous lemmas and conclude that the expected training error converges at a linear rate through Lemma A.7:

$$\begin{aligned}
\frac{d}{dt}R_\mathcal{S}(t) &= \frac{1}{2}\frac{d}{dt}\left\|\hat{f}(\mathbf{X};t) - \mathbf{y}\right\|_2^2 = -\left(\mathbf{y} - \hat{f}(\mathbf{X},t)\right)^\top \left(\boldsymbol{\Theta}_\mu(t) + \boldsymbol{\Theta}_\sigma(t)\right)\left(\mathbf{y} - \hat{f}(\mathbf{X},t)\right) \\
&\le -\lambda_0\left(\boldsymbol{\Theta}_\mu(t) + \boldsymbol{\Theta}_\sigma(t)\right)\left\|\mathbf{y} - \hat{f}(\mathbf{X};t)\right\|_2^2 \le -\lambda_0\left(\boldsymbol{\Theta}_\mu^{(L)}(t) + \boldsymbol{\Theta}_\sigma^{(L)}(t)\right)\left\|\mathbf{y} - \hat{f}(\mathbf{X};t)\right\|_2^2 \\
&\le -\lambda_0\left(\mathbf{K}_\infty^{(L)}\right)\left\|\mathbf{y} - \hat{f}(\mathbf{X};t)\right\|_2^2.
\end{aligned}$$

Note our proof framework follows Du et al. (2019)'s, especially, we share the same three core steps. However, the main difference is that our network architecture is much more complex. Because the probabilistic network contains two sets of parameters, we have two NTKs. As a result, the proof requires bounding many terms more elaborately. For example, in Lemma A.3 and Lemma A.5 we bound the PNTK associated with variance. The detailed proof can be found in Appendix A.

## 5.2 Proof of Theorem 4.3

The proof of Theorem 4.3 utilizes an argument of linearization of the network model in the infinite-width limit. This allows us to obtain an ordinary differential equation for output function.

According the linearization rules for infinitely-wide networks (Lee et al., 2019), the output function can be expressed as,

$$\hat{f}^\infty(\mathbf{x}, t) = \boldsymbol{\phi}_\mu(\mathbf{x})^\top(\boldsymbol{\theta}_\mu(t) - \boldsymbol{\theta}_\mu(0)) + \boldsymbol{\phi}_\sigma(\mathbf{x})^\top(\boldsymbol{\theta}_\sigma(t) - \boldsymbol{\theta}_\sigma(0)),$$

where $\boldsymbol{\phi}_\mu(\mathbf{x}) = \nabla_{\boldsymbol{\theta}_\mu}\hat{f}(\mathbf{x};0)$, and $\boldsymbol{\phi}_\sigma(\mathbf{x}) = \nabla_{\boldsymbol{\theta}_\sigma}\hat{f}(\mathbf{x};0)$. The next step is to show the difference between finite-width neural network and infinitely-wide network:

$$\left|\hat{f}(\mathbf{x}_{te}) - \hat{f}^\infty(\mathbf{x}_{te})\right| \leq O(\mathcal{E}),$$

where $\mathcal{E} = \mathcal{E}_{\text{init}} + \frac{\sqrt{n}\mathcal{E}_\Theta}{\lambda_0 + \beta}$ with $\left\|\hat{f}(\mathbf{x}_{te};0)\right\|_2 \leq \mathcal{E}_{\text{init}}$ and $\|\boldsymbol{\Theta}_\mu^\infty - \boldsymbol{\Theta}_\mu(t)\|_2 \leq \mathcal{E}_\Theta$. The details to complete the proof are given in Appendix B.

## 6 Experiments

As an extension of our finding of the PAC-Bayesian bound in Theorem 4.4, we present a training-free metric to approximate the PAC-Bayesian bound using PNTK, which can be used to select the best hyper-parameters without the need for additional training and eliminate excessive computation time. Besides, we provide an empirical verification of our theory in Appendix C.2 to further demonstrate the correctness of theoretical analysis.

### 6.1 Experimental setup

In all experiments, we initialize the PNN parameters using the NTK parameterization as given in Equation (1). The initial mean weights $\boldsymbol{\theta}_\mu$ are sampled from a truncated Gaussian distribution with mean 0 and standard deviation 1, truncated at two standard deviations. To ensure positivity, the initial variance weights $\boldsymbol{\theta}_\sigma$ are set by transforming the specified value of $c_\sigma$ through the formula $c_\sigma = \log(1 + \exp(\rho_0))$.

In Section 6.2, we conduct experiments on the MNIST and CIFAR-10 datasets to demonstrate the effectiveness of our training-free PAC-Bayesian network bound for hyperparameter search under different network architectures. We consider both fully connected and convolutional neural networks, with a 3-layer fully connected network with 600 neurons on each layer and a 13-layer convolutional architecture with around 10 million learnable parameters. We adopt a data-dependent prior, which is a practical and popular method (Dziugaite et al., 2021; Perez-Ortiz et al., 2021; Fortuin, 2022), pre-trained on a subset of the total training data with empirical risk minimization, and the networks for posterior training are initialized by the weights learned from the prior. To compute the generalization bound, we use Equation (5), with the relevant settings from the work by Pérez-Ortiz et al. (2021), such as confidence $\delta$ for the risk certificate (5) and 150,000 Monte Carlo samples to estimate the risk certificate.

### 6.2 Selecting hyperparameters via training-free metric

The PAC-Bayesian learning framework provides competitive performance with non-vacuous generalization bounds. However, the tightness of this generalization bounds heavily depends on the hyperparameters used, such as the proportionality of data used for the prior, the initialization of $\rho_0$, and the KL penalty weight ($\lambda$). While it is worth noting that there is a "penalty" associated with $\delta$ (see Equation 5), this penalty is typically relatively small compared to the potential performance improvements that can be gained through optimal hyperparameter selection. As these values remain fixed during the training process, selecting optimal hyperparameters is a critical and computationally challenging task. Grid search is one possible approach, but it can be prohibitively expensive due to the significant computational resources needed to compute the generalization bounds for each combination of hyperparameters.

To this end, we propose an approach using "training-free" metric to approximate the generalization bound without performing a time-consuming training process based on a generalization bound developed in theorem 4.4 via NTK. As PNTK is constant during training, we use an upper bound of Equation 7 with PNTK by as a proxy metric, which can be formulated as follows:

$$\mathcal{PA} = \text{Tr}\left(\frac{(\widehat{\boldsymbol{\Theta}} + \lambda/c_\sigma^2\mathbf{I})^{-1} \cdot \mathbf{y}\mathbf{y}^\top}{c_\sigma^2 \cdot n} + \frac{\lambda}{c_\sigma^2}\sqrt{\frac{(\widehat{\boldsymbol{\Theta}} + \lambda/c_\sigma^2\mathbf{I})^{-2} \cdot \mathbf{y}\mathbf{y}^\top}{n}}\right), \tag{20}$$

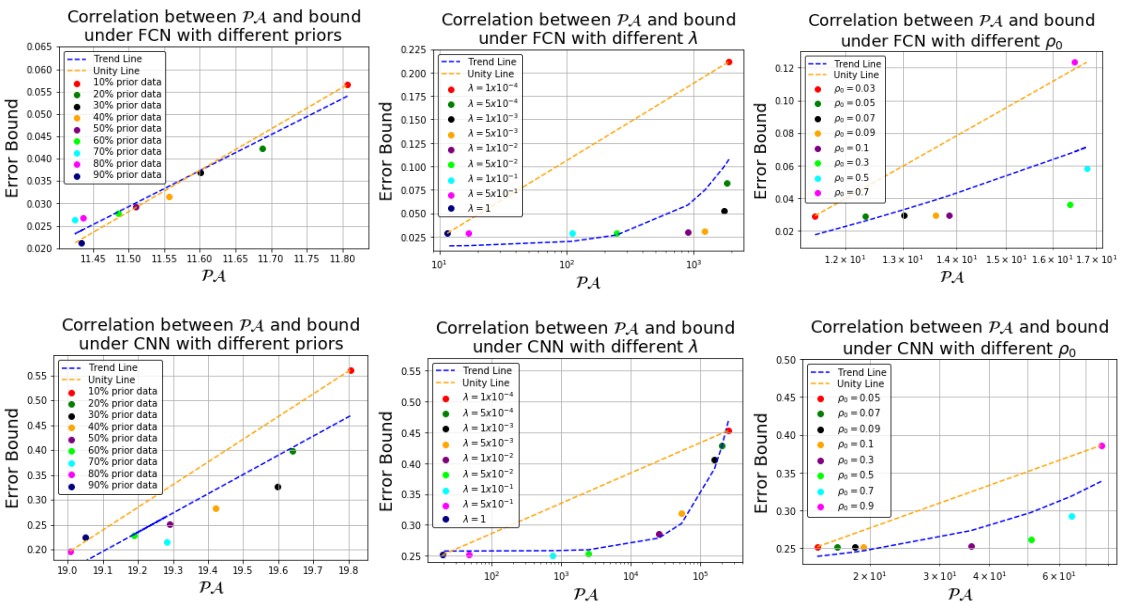

Figure 1: The first row shows correlation results between the generalization bound and the proportion of prior data, the coefficient of the KL penalty, and $\rho_0$ for the FCN structure on the MNIST dataset. To better visualize the trend, we have implemented a trend line, which represents the overall trend of a set of data points, and a 1-to-1 line, which represents a perfect relationship between two variables where the values on the x-axis are equal to the values on the y-axis. We find high Kendall-tau correlations of 0.89, 0.89, and 0.93. Similar results are obtained in the second row for the CNN structure on the CIFAR10 dataset, where the Kendall-tau correlations are 0.89, 0.83, and 0.57, respectively.

where $\widehat{\boldsymbol{\Theta}}$ is an empirical NTK measured on a finite-width neural network at initialization, and $\mathbf{y}\mathbf{y}^{\top}$ is a $n \times n$ label similarity matrix, where a joint entry is one if two data points have the same label, and zero otherwise. We leave the derivation process in Appendix C.1 and note that the proposed proxy metric in Equation (20) is similar in spirit to kernel alignment, a label similarity metric, that is widely used in deep active learning (Wang et al., 2022a), model selection for fine-tuning (Deshpande et al., 2021), and neural architecture search (NAS) (Mok et al., 2022). We should also mention that training-free methods for searching neural architectures are not new, and can be found in NAS (Chen et al., 2021; Deshpande et al., 2021), MAE Random Sampling (Camero et al., 2021), pruning at initialization (Abdelfattah et al., 2021). Further more both Yang et al. (2021) and Immer et al. (2021) focus on hyperparameter (HP) tuning and model selection. In contrast to both methods, which involve training, our approach is training-free. We have included a comparison in the updated manuscript. To the best of our knowledge, this is the first training-free method for selecting hyperparameters in the PAC-Bayesian framework, which we consider to be one of the novelties of this paper.

To demonstrate the computational practicality of this training-free metric, we compute $\mathcal{PA}$ using a subset of the data for each class (325 per class for FCN and 75 per class for CNN). Figure 1 demonstrates a strong correlation between $\mathcal{PA}$ and the actual generalization bound. Importantly, we show that using $\mathcal{PA}$ to search for hyperparameters can yield a result that is comparable to the best generalization bound, but with significantly reduced computational time. We compare the performance of three hyperparameter search methods (exhaustive search, Bayesian search, and $\mathcal{PA}$) on two architectures (FCN and CNN) and two datasets (MNIST and CIFAR10) in Table 1. Exhaustive search evaluates 648 hyperparameter combinations (9 data-dependent prior with different subsets data for prior training, 9 different values of KL penalty, and 8 different values of $\rho_0$), while Bayesian search takes only 36 iterations to find the lowest bound, but still requires significant computation time when training a large and complex model. For instance, under the CIFAR10 dataset, it takes 45 hours to train a CNN with the bound. In contrast, using $\mathcal{PA}$ saves 83.33

| Setup | | | Risk cert. | | Test err. | | Computation time (hours) | |
|---|---|---|---|---|---|---|---|---|
| Data | Method | Network | $\ell^{\text{x-e}}$ | $\ell^{01}$ | x-e | acc. | Single | Total |
| MNIST | Exhaustive Search | FCN | .0010 | .0212 | .0001 | .0189 | .50 | 324.00 |
| | | CNN | **.0006** | **.0110** | .0004 | .0093 | 16.92 | 10964.16 |
| | Bayesian Search | FCN | .0010 | .0212 | .0001 | .0189 | .50 | **18.00** |
| | | CNN | **.0006** | **.0110** | .0004 | .0093 | 16.92 | 609.12 |
| | $\mathcal{PA}$ | FCN | .0011 | .0264 | .0002 | .0208 | .03 | 19.44 |
| | | CNN | .0009 | .0160 | .0008 | .0108 | .03 | 19.44 |
| CIFAR10 | Exhaustive Search | FCN | .1740 | .5377 | .0051 | .4866 | 1.09 | 706.32 |
| | | CNN | **.0142** | **.1969** | .0023 | .1510 | 45.00 | 29,160.00 |
| | Bayesian Search | FCN | .1740 | .5377 | .0051 | .4866 | 1.09 | 39.24 |
| | | CNN | **.0142** | **.1969** | .0023 | .1510 | 45.00 | 1,620.00 |
| | $\mathcal{PA}$ | FCN | .1780 | .5490 | .0048 | .4920 | .03 | **19.44** |
| | | CNN | **.0142** | .1970 | .0024 | .1511 | .03 | **19.44** |

Table 1: The performance of our training-free method, $\mathcal{PA}$, is compared with two other hyperparameter search methods, exhaustive search and Bayesian search, on two datasets (MNIST and CIFAR10) and two neural network structures (FCN and CNN). The evaluation is based on the risk certificates (cross-entropy $\ell^{\text{x-e}}$ and accuracy $\ell^{01}$, test error (cross-entropy x-e and accuracy acc.), and computation time. The best and second-best values of risk certificates and computation time are highlighted in boldface and underlining, respectively.

times the computational time to find a bound that is close to the lowest risk certificate in accuracy. The results show that the $\mathcal{PA}$ method achieves comparable performance to the other methods while requiring significantly less computation time.

## 7 Conclusion and Discussion

In this work, we have made several important contributions to the theoretical analysis and practical implementation of deep probabilistic neural networks trained using objectives derived from PAC-Bayes bounds. Specifically, we have shown that the learning dynamics of these networks in an over-parameterized setting can be exactly described by the PNTK, and we have confirmed this through empirical investigation. We have also demonstrated that the expected output function trained with a PAC-Bayesian bound converges to the kernel ridge regression, leading to an explicit generalization bound. Moreover, we have proposed a training-free method, which can effectively select hyper-parameters and lead to lower generalization bounds without the excessive computational cost of brute-force grid search. Our work opens up several promising directions for future research. One such direction is the study of PAC-Bayesian learning with data-dependent priors using PNTK. We hope that our contributions will help advance the understanding of Deep PAC-Bayesian Learning and implementation of deep probabilistic neural networks and their potential applications.

### Acknowledgments

We express our gratitude to the anonymous reviewers and Action Editors, whose insightful suggestions significantly enhanced the quality of our paper. Our special thanks go to Tianyu Liu, whose constructive conversations during the preliminary stages of this project were invaluable.

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

# A Proof of Theorem 4.2

**Theorem A.1** (Restatement of Theorem 4.2). *Suppose $\sigma(\cdot)$ is $H$-Lipschitz, $\beta$-Smooth. Assume $\sigma(\cdot)$ and its partial derivative $\frac{\partial \sigma(x)}{\partial x}$ are continuous in $x$, $\lambda_0(\mathbf{K}_\infty^{(L)}) > 0$, and the network's width is of $m = \Omega\left(2^{O(L)} \max\left\{\frac{n^2 \log(Ln/\delta)}{\lambda_0^2(\mathbf{K}_\infty^{(L)})}, \frac{n}{\delta}, \frac{n^5 \log(2/\delta)^{10}}{\lambda_0^2(\mathbf{K}_\infty^{(L)})}\right\}\right)$. Then, with a probability of at least $1 - \delta$ over the random initialization, we have,*

$$R_{\mathcal{S}}(Q(t)) \leq \exp\left(-\lambda_0(\mathbf{K}_\infty^{(L)})t\right) R_{\mathcal{S}}(Q(0)),$$

*Proof Sketch of Theorem A.1.* To study the behavior of output function under gradient flow, we first write down its dynamics

$$\frac{d\hat{f}(\mathbf{X}; t)}{dt} = \frac{1}{n}(\mathbf{y} - \hat{f}(\mathbf{X}; t))(\mathbf{\Theta}_\mu(\mathbf{X}, \mathbf{X}; t) + \mathbf{\Theta}_\sigma(\mathbf{X}, \mathbf{X}; t)),$$

where $\mathbf{\Theta}_\mu$ and $\mathbf{\Theta}_\sigma$ are the PNTKs of the PNN. We observe that if $\mathbf{\Theta}_\mu$ and $\mathbf{\Theta}_\sigma$ converge to a deterministic kernel, then the dynamics of output function admit a linear system, which is tractable during evolution. In this paper we focus on $\mathbf{\Theta}^{(L)} = \mathbf{\Theta}_\mu^{(L)} + \mathbf{\Theta}_\sigma^{(L)}$, the gram matrix induced by the weights from $L$-th layer for simplicity at the cost of a minor degradation in convergence rate. Before demonstrating the main steps, we introduce the expected neural network model, which is recursively expressed as follows:

$$\hat{\mathbf{x}}^{(l)} = \mathbb{E}_{\boldsymbol{\xi}^{(l)}} \frac{1}{\sqrt{m}} \sigma(\mathbf{W}^{(l)} \hat{\mathbf{x}}^{(l-1)}), \quad 1 \leq l \leq L-1; \quad \hat{f} = \mathbb{E}_{\boldsymbol{\xi}^{(L)}} \frac{1}{\sqrt{m}} \mathbf{W}^{(L)} \hat{\mathbf{x}}^{(L-1)}.$$

Note that this definition is in inordinate to the the definition as expected empirical loss (9) in PAC-Bayes. a Neural Network Gaussian Process (NNGP) for PNN (Lee et al., 2018), which is defined as follows:

$$K_{ij}^{(l)} = (\hat{\mathbf{x}}_i^{(l)})^\top \hat{\mathbf{x}}_j^{(l)}, \quad K_{ij,\infty}^{(l)} = \lim_{m \to \infty} K_{ij}^{(l)},$$

where subscript $i, j$ denote the index of input samples. We aim to show that $\mathbf{\Theta}^{(L)}$ is close to $\mathbf{K}_\infty^{(L)}$ in large width. In particular, to prove Theorem A.1, three core steps are:

Step 1 Show at initialization $\lambda_0(\mathbf{\Theta}^{(L)}(0)) \geq \frac{\lambda_0(\mathbf{K}_\infty^{(L)})}{2}$ and the required condition on $m$.

Step 2 Show during training $\lambda_0(\mathbf{\Theta}^{(L)}(t)) \geq \frac{\lambda_0(\mathbf{K}_\infty^{(L)})}{2}$ and the required condition on $m$.

Step 3 Show during training the expected empirical loss $R_{\mathcal{S}}(Q)$ has a linear convergence rate.

In our proof, we mainly focus on deriving the condition on $m$ by analyzing $\lambda_0(\mathbf{\Theta}^{(L)}(0))$ at initialization through Lemma A.2 and Lemma A.3. For step 2, we construct Lemma A.4 Lemma A.5 and Lemma A.6 to demonstrate that $\lambda_0(\mathbf{\Theta}^{(L)}(t)) \geq \frac{\lambda_0(\mathbf{K}_\infty^{(L)})}{2}$. This leads to the required condition on $m$ during train. Finally, we summarize all the previous lemmas and conclude that the expected training error converges at a linear rate through Lemma A.7.

$\square$

## A.1 Step 1. Bounding least eigenvalue of PNTK at initialization

We first study the behavior of tangent kernels with an ultra-wide condition, namely $m = \text{poly}(n, 1/\lambda_0, 1/\delta)$ at initialization. Lemmas A.2 and A.3 demonstrate that if $m$ is large, then the feature of each layer is approximately normalized, $\mathbf{\Theta}_\mu(0) + \mathbf{\Theta}_\sigma(0)$ have a lower bound on the smallest eigenvalue with a high probability.

**Lemma A.2** (Initial norm at initialization)**.** *Suppose $\sigma(\cdot)$ is H-Lipschitz. If $m = \Omega\left(\frac{nLg_C(L)^2}{\delta}\right)$, where $C \triangleq (c_\mu^2 + c_\sigma^2)H(2|\sigma(0)|\sqrt{\frac{2}{\pi}} + 2H)$, and the geometric series function $g_C(l) = \sum_{i=0}^{l-1} C^i$. Then with probability at least $1 - \delta$ over random initialization, for each $l \in [L]$ and $i \in [n]$, we have:*

$$\frac{1}{2} \leq \|\hat{\mathbf{x}}_i^{(l)}(0)\|_2 \leq 2.$$

**Lemma A.3** (PNTK at initialization)**.** *Suppose $\sigma(\cdot)$ is H-Lipschitz. If $m = \Omega\left(\frac{n^2 \log(Ln/\delta)2^{O(L)}}{\lambda_0^2(\mathbf{K}_\infty^{(L)})}\right)$, then with probability at least $1 - \delta$, we have:*

$$\lambda_0(\mathbf{\Theta}^{(L)}(0)) \geq \frac{3}{4}\lambda_0(\mathbf{K}_\infty^{(L)}).$$

*Proof of Lemma A.2.* The proof is by induction method. The induction hypothesis is that with probability at least $1 - (l-1)\frac{\delta}{nL}$ over $\mathbf{W}_\mu^{(1)}(0), \ldots, \mathbf{W}_\mu^{(l-1)}(0)$, for every $1 \leq l' \leq l - 1$, we have

$$\frac{1}{2} \leq 1 - \frac{g_C(l')}{2g_C(L)} \leq \|\hat{\mathbf{x}}_i^{(l')}(0)\|_2 \leq 1 + \frac{g_C(l')}{2g_C(L)} \leq 2,$$

where we define the geometric series function as $g_C(l) = \sum_{i=0}^{l-1} C^i$.

According to the feed-forward expression, we know that

$$\|\hat{\mathbf{x}}_i^{(l)}(0)\|_2^2 = \frac{1}{m} \sum_{r=1}^m \mathbb{E}_{\boldsymbol{\xi}^{(l)}} \sigma((\mathbf{w}_{\mu,r}^{(l)}(0) + c_\sigma^2\boldsymbol{\xi}^{(l)})^\top \hat{\mathbf{x}}_i^{l-1}(0))^2.$$

Then we have the expectation:

$$\mathbb{E}\left[\|\hat{\mathbf{x}}_i^{(l)}(0)\|_2^2\right] = \mathbb{E}_{\mathbf{w},\boldsymbol{\xi}^{(l)}}\left[\sigma((\mathbf{w}^{(l)} + c_\sigma^2\boldsymbol{\xi}^{(l)})^\top \hat{\mathbf{x}}_i^{(l-1)}(0))^2\right]$$

$$= (c_\mu^2 + c_\sigma^2)\mathbb{E}_{Z\sim\mathcal{N}(0,1)}\sigma(\|\hat{\mathbf{x}}^{(l-1)}\|_2 Z)^2.$$

Here we have used the fact that $\mathbf{w}^{(l)}(0) + c_\sigma^2\boldsymbol{\xi}^{(l)} \sim (c_\mu^2 + c_\sigma^2)\mathcal{N}(\mathbf{0}, \mathbf{I})$ where $\mathbf{w}^{(l)}(0) \sim c_\mu^2\mathcal{N}(\mathbf{0}, \mathbf{I})$.

Because $\sigma(\cdot)$ is $H$-Lipschitz, for $\frac{1}{2} \leq \alpha \leq 2$, we have

$$\left|\mathbb{E}_{Z\sim\mathcal{N}(0,1)}\left[\sigma(\alpha Z)^2\right] - \mathbb{E}_{Z\sim\mathcal{N}(0,1)}\left[\sigma(Z)^2\right]\right|$$

$$\leq \mathbb{E}_{Z\sim\mathcal{N}(0,1)}\left[|\sigma(\alpha Z)^2 - \sigma(Z)^2|\right]$$

$$\leq H|\alpha - 1| \cdot \mathbb{E}_{Z\sim\mathcal{N}(0,1)}\left[|Z(\sigma(\alpha Z) + \sigma(Z))|\right]$$

$$\leq H|\alpha - 1| \cdot \mathbb{E}_{Z\sim\mathcal{N}(0,1)}\left[|Z|(|2\sigma(0)| + H|(\alpha + 1)Z|)\right]$$

$$\leq H|\alpha - 1| \cdot (2|\sigma(0)|\mathbb{E}_{Z\sim\mathcal{N}(0,1)}[|Z|] + H|\alpha + 1| \cdot \mathbb{E}_{Z\sim\mathcal{N}(0,1)}[Z^2])$$

$$= H|\alpha - 1| \cdot (2|\sigma(0)|\sqrt{\frac{2}{\pi}} + H|\alpha + 1|)$$

$$\leq \frac{C}{c_\mu^2 + c_\sigma^2}|\alpha - 1|,$$

where we define $C \triangleq (c_\mu^2 + c_\sigma^2)H(2\sigma(0)\sqrt{\frac{2}{\pi}} + 2H)$. For the variance we have:

$$\text{Var}\left[\|\hat{\mathbf{x}}_i^{(l)}(0)\|_2^2\right] = \text{Var}\left[\sigma(\mathbf{w}_r^{(l)}(0)^\top \hat{\mathbf{x}}_i^{(l)}(0))^2\right]$$

$$\leq \frac{1}{m}\mathbb{E}_{\mathbf{w}^{(l)},\boldsymbol{\xi}^{(l)}}\left[\sigma((\mathbf{w}^{(l)} + c_\sigma^2\boldsymbol{\xi}^{(l)})^\top \hat{\mathbf{x}}_i^{(l)}(0))^4\right]$$

$$\leq \frac{(c_\sigma^2 + c_\mu^2)^2}{m}\mathbb{E}_{\mathbf{w}\sim\mathcal{N}(\mathbf{0},\mathbf{I})}\left[\left(|\sigma(0)| + H|\mathbf{w}^\top\hat{\mathbf{x}}_i^{(l)}(0)|\right)^4\right]$$

$$\leq \frac{C_2}{m},$$

where $C_2 \triangleq (c_\sigma^2 + c_\mu^2)^2 (\sigma(0)^4 + 8|\sigma(0)|^3 H \sqrt{2/\pi} + 24\sigma(0)^2 H^2 + 64\sigma(0)H^3\sqrt{2/\pi} + 512H^4)$, and the last inequality we used the formula for the first four absolute moments of Gaussian.

Applying Chebyshev's inequality and plugging in our assumption on $m$, we have with probability $1 - \frac{\delta}{nL}$ over $\mathbf{W}_\mu^{(l)}$,

$$\left| \left\| \hat{\mathbf{x}}_i^{(l)}(0) \right\|_2^2 - \mathbb{E}\left\| \hat{\mathbf{x}}_i^{(l)}(0) \right\|_2^2 \right| \leq \frac{1}{2g_C(L)}.$$

Thus with probability $1 - d\frac{\delta}{nL}$ over $\mathbf{W}_\mu^{(1)}, \ldots, \mathbf{W}_\mu^{(l)}$,

$$\begin{aligned}
\left| \left\| \hat{\mathbf{x}}_i^{(l)}(0) \right\|_2 - 1 \right| &\leq \left| \left\| \hat{\mathbf{x}}_i^{(l)}(0) \right\|_2^2 - 1 \right| \\
&\leq \frac{Cg_C(l-1)}{2g_C(L)} + \frac{1}{2g(L)} \\
&= \frac{g_C(l)}{2g_C(L)}.
\end{aligned}$$

Using union bounds over $i \in [n]$, we complete the proof of lemma. $\qquad\square$

*Proof of Lemma A.3.* For a weight matrix, we decompose it into $m$ weight vectors, namely $\mathbf{W}^{(l)} = [\mathbf{w}_1^{(l)}, \mathbf{w}_2^{(l)}, \cdots, \mathbf{w}_m^{(l)}]$. Then the derivative of the expected output over the parameters $\mathbf{w}_{\mu,r}^{(l)}$ and $\mathbf{w}_{\sigma,r}^{(l)}$ can be expressed as

$$\begin{aligned}
\frac{\partial \hat{f}(\mathbf{x}_i)}{\partial \mathbf{w}_{\mu,r}^{(l)}} &= \mathbb{E}_{\boldsymbol{\xi}}\left[ \frac{1}{\sqrt{m}} \frac{\partial \hat{f}(\mathbf{x}_i)}{\partial \hat{x}_{i,r}^{(l)}} \sigma'((\mathbf{w}_r^{(l)})^\top \hat{\mathbf{x}}^{(l-1)}) \hat{\mathbf{x}}^{(l-1)} \right], \\
\frac{\partial \hat{f}(\mathbf{x}_i)}{\partial \mathbf{w}_{\sigma,r}^{(l)}} &= \mathbb{E}_{\boldsymbol{\xi}}\left[ \frac{1}{\sqrt{m}} \frac{\partial \hat{f}(\mathbf{x}_i)}{\partial \hat{x}_{i,r}^{(l)}} \sigma'((\mathbf{w}_r^{(l)})^\top \hat{\mathbf{x}}^{(l-1)}) \hat{\mathbf{x}}^{(l-1)} \odot \boldsymbol{\xi}_r^{(l)} \right],
\end{aligned} \tag{21}$$

where we have interchanged integration and differentiation over activation $\sigma(\cdot)$. According to the definition 4.1 of PNTK for each layer can be expressed as:

$$\boldsymbol{\Theta}_\mu^{(l)} = \sum_{r=1}^m \nabla_{\mathbf{w}_{\mu,r}^{(l)}} \hat{f}(\mathbf{X}; t)^\top \nabla_{\mathbf{w}_{\mu,r}^{(l)}} \hat{f}(\mathbf{X}; t), \quad \boldsymbol{\Theta}_\sigma^{(l)} = \sum_{r=1}^m \nabla_{\mathbf{w}_{\sigma,r}^{(l)}} \hat{f}(\mathbf{X}; t)^\top \nabla_{\mathbf{w}_{\sigma,r}^{(l)}} \hat{f}(\mathbf{X}; t).$$

Through a standard calculation we show that:

$$\boldsymbol{\Theta}_{\mu,ij}^{(L)}(0) = \sum_{r=1}^m \mathbb{E}_{\boldsymbol{\xi}^{(L-1)}}\left[ \frac{1}{\sqrt{m}} \sigma((\mathbf{w}_r^{(L-1)})^\top \hat{\mathbf{x}}_i^{(L-2)}) \right] \mathbb{E}_{\boldsymbol{\xi}^{(L-1)}}\left[ \sigma((\mathbf{w}_r^{(L-1)})^\top \hat{\mathbf{x}}_j^{(L-2)}) \right], \tag{22}$$

$$\boldsymbol{\Theta}_{\sigma,ij}^{(L)}(0) = \sum_{r=1}^m \mathbb{E}_{\boldsymbol{\xi}^{(L-1)}, \xi_r^{(L)}}\left[ \frac{1}{\sqrt{m}} \sigma((\mathbf{w}_r^{(L-1)})^\top \hat{\mathbf{x}}_i^{(L-2)})\xi_r^{(L)} \right] \mathbb{E}_{\boldsymbol{\xi}^{(L-1)}, \xi_r^{(L)}}\left[ \frac{1}{\sqrt{m}} \sigma((\mathbf{w}_r^{(L-1)})^\top \hat{\mathbf{x}}_j^{(L-2)})\xi_r^{(L)} \right]. \tag{23}$$

**Bounding $\boldsymbol{\Theta}_\mu^{(L)}$.** The proof is by induction. By analyzing Equation 22, we find that for all pairs of $i, j$, $\boldsymbol{\Theta}_{\mu,ij}^{(L)}(0)$ is the average of $m$ i.i.d. random variables, with the expectation:

$$\mathbf{K}_{ij,\infty}^{(L)} = c_\mu^2 \cdot \mathbb{E}_{\mathbf{w}\sim\mathcal{N}(\mathbf{0},\mathbf{I})}\left[ \mathbb{E}_{\boldsymbol{\xi}}\left[ \frac{1}{\sqrt{m}} \sigma(\mathbf{w} + c_\sigma^2\boldsymbol{\xi})^\top \hat{\mathbf{x}}^{(L-1)}) \right] \mathbb{E}_{\boldsymbol{\xi}}\left[ \frac{1}{\sqrt{m}} \sigma((\mathbf{w} + c_\sigma^2\boldsymbol{\xi})^\top \hat{\mathbf{x}}^{(L-1)}) \right] \right].$$

Then we find the difference between $\boldsymbol{\Theta}_{\mu,ij}^{(L)}(0)$ and $\mathbf{K}_{ij,\infty}^{(L)}$ can be recursively calculated through an induction method. The induction hypothesis is that with probability $1 - \delta$ over the $\left\{ \mathbf{W}_\mu^{(l)} \right\}_{l=1}^{L-1}$, for any $1 \leq l \leq L - 1, 1 \leq i, j \leq n$,

$$\left\| \frac{1}{m} \sum_{r=1}^m (\hat{\mathbf{x}}_{i,r}^{(l)})^\top \hat{\mathbf{x}}_{j,r}^{(l)} - K_{ij}^{(l)} \right\|_\infty \leq C^L \sqrt{\frac{\log(Ln/\delta)}{m}}, \tag{24}$$

where $C$ is a constant.

We start from the first layer:

$$\mathbb{E}\left[\frac{1}{m}\sum_{r=1}^{m}\hat{\mathbf{x}}_{i,r}^{(1)}\hat{\mathbf{x}}_{j,r}^{(1)\top}\right] = K_{ij,\infty}^{(1)}.$$

We can apply standar Hoeffding bound and obtain the following concentration inequalities. With probability at least $1 - \frac{\delta}{L}$, we have

$$\max_{i,j}\left\|\frac{1}{m}\sum_{i=1}^{m}\hat{\mathbf{x}}_{i}^{(1)}\hat{\mathbf{x}}_{j}^{(1)\top} - K_{ij,\infty}^{(1)}\right\|_{\infty} \leq \sqrt{\frac{2\log(4Ln^2)^2/\delta)}{m}}.$$

Now we prove the induction step. In the following, by $\mathbb{E}^{(l)}$ we mean taking expectation over $\mathbf{w}^{(l)} \sim \mathcal{N}(\mathbf{0}, c_\mu^2\mathbf{I})$. Suppose that Equation (24) holds for $1 \leq l' \leq l$ with probability at least $1 - \frac{l}{L}\delta$, then we want to show the equations holds for $l + 1$ with probability at least $1 - \delta/L$ conditioned on previous layers satisfying Equation 24. Same as the base case, applying concentration inequalities, we have with probability at least $1 - \delta/L$,

$$\max_{ij}\left\|\mathbb{E}^{(l)}K_{ij}^{(l)} - K_{ij}^{(l)}\right\|_{\infty} \leq \sqrt{\frac{2\log(4Ln^2)^2/\delta)}{m}}.$$

Now it remains to bound the differences:

$$\begin{aligned}
\max_{ij}\left\|\mathbb{E}^{(l)}K_{ij}^{(l)} - K_{ij,\infty}^{(l)}\right\|_{\infty} &\leq \left\|\mathbb{E}_{(u^{(l)},v^{(l)})\sim\mathbf{A}}\mathbb{E}_Z\sigma(u^{(l)} + c_\sigma^2 Z)\mathbb{E}_Z\sigma(v^{(l)} + c_\sigma^2 Z)\right. \\
&\qquad - \left.\mathbb{E}_{(u^{(l)},v^{(l)})\sim\mathbf{A}_\infty}\mathbb{E}_Z\sigma(u^{(l)} + c_\sigma^2 Z)\mathbb{E}_Z\sigma(v^{(l)} + c_\sigma^2 Z)\right\| \\
&\overset{(a)}{\leq} C\|\mathbf{A} - \mathbf{A}_\infty\|_F \\
&\leq 2C\|\mathbf{A} - \mathbf{A}_\infty\|_\infty \\
&\leq C_1\max_{ij}\|K_{ij}^{(l-1)} - K_{ij,\infty}^{(l-1)}\|_\infty,
\end{aligned}$$

where $\mathbf{A} = \begin{bmatrix} K_{ii}^{(l-1)} & K_{ij}^{(l-1)} \\ K_{ji}^{(l-1)} & K_{jj}^{(l-1)} \end{bmatrix}$ and $\mathbf{A}_\infty = \begin{bmatrix} K_{ii,\infty}^{(l-1)} & K_{ij,\infty}^{(l-1)} \\ K_{ji,\infty}^{(l-1)} & K_{jj,\infty}^{(l-1)} \end{bmatrix}$, (a) is by property of activation function and Taylor's Theorem, and $C_1$ and $C_2$ are positive constants.

Thus by matrix perturbation theory we have,

$$\left\|\mathbf{\Theta}_\mu^{(L)} - \mathbf{K}_\infty^{(l)}\right\|_2 \leq \sum_{ij}\left|\mathbf{\Theta}_{\mu,ij}^{(L)} - \mathbf{K}_{ij,\infty}^{(l)}\right| \leq C^L n\sqrt{\frac{\log(Ln/\delta)}{m}}.$$

**Bounding $\mathbf{\Theta}_\sigma^{(L)}(0)$.** By analyzing Equation (23), the independent random variable $\xi_r^{(L)}$ yields:

$$\mathbf{\Theta}_{\sigma,ij}^{(L)}(0) = \sum_{r=1}^{m}\mathbb{E}_{\boldsymbol{\xi},\xi_r^{(L)}}\left[\frac{1}{\sqrt{m}}\sigma((\mathbf{w}_r^{(L-1)})^\top\mathbf{x}_i^{(L-2)})\xi_r^{(L)}\right]\mathbb{E}_{\boldsymbol{\xi},\xi_r^{(L)}}\left[\frac{1}{\sqrt{m}}\sigma((\mathbf{w}_r^{(L-1)})^\top\mathbf{x}_j^{(L-2)})\xi_r^{(L)}\right] = 0.$$

Therefore, at initialization and during, the $\mathbf{\Theta}_\sigma^{(L)}(0)$ keeps zero.

Finally, if $m = \Omega\left(\frac{n^2\log(Ln/\delta)2^{O(L)}}{\lambda_0^2(\mathbf{K}_\infty^{(L)})}\right)$, then with probability at least $1 - \delta$,

$$\left\|\mathbf{\Theta}^{(L)}(0) - \mathbf{K}_\infty^{(L)}\right\|_2 = \left\|\mathbf{\Theta}_\mu^{(L)}(0) - \mathbf{K}_\infty^{(L)}\right\|_2 \leq \frac{\lambda_0(\mathbf{K}^{(L)})}{4},$$

which completes the proof. $\qquad\square$

### A.2   Step 2. Bounding least eigenvalue of PNTK during training.

The next problem is that PNTKs are time-dependent matrices, thus varying during training. To account for this problem, we establish following lemmas stating that if the weights during training are close to their initialization, then the corresponding PNTK is close to their initialization.

**Lemma A.4.** *Suppose for every* $l \in [L]$, $\left\|\mathbf{W}_\mu^{(l)}(0)\right\|_2 \leq c_{\mu,0}\sqrt{m}$, $\left\|\hat{\mathbf{x}}^{(l)}(0)\right\|_2 \leq c_{x,0}$ *and* $\left\|\mathbf{W}_\mu^{(l)}(t) - \mathbf{W}_\mu^{(l)}(0)\right\|_F \leq \sqrt{m}R$ *for some constant* $c_{\mu,0}, c_{x,0} > 0$ *and* $R \leq c_{\mu,0}$. *If* $\sigma(\cdot)$ *is* $H$-Lipschitz, then with probability at least $1 - \delta$, we have

$$\left\|\hat{\mathbf{x}}^{(l)}(t) - \hat{\mathbf{x}}^{(l)}(0)\right\|_2 \leq c_{x,0}HRg_{c_x}(l),$$

*where* $c_x = 2Hc_{\mu,0}$.

**Lemma A.5.** *Suppose* $\sigma(\cdot)$ *is* $H-$*Lipschitz and* $\beta-$*smooth. Suppose for* $l \in [L]$, $\left\|\mathbf{W}_\mu^{(l)}(0)\right\|_2 \leq c_{\mu,0}\sqrt{m}$, $\frac{1}{c_{x,0}} \leq \left\|\hat{\mathbf{x}}^{(l)}(0)\right\|_2 \leq c_{x,0}$. *If* $\left\|\mathbf{W}_\mu^{(l)}(t) - \mathbf{W}_\mu^{(l)}(0)\right\|_F \leq \sqrt{m}R$ *where* $R \leq cg_{c_x}(L)^{-1}\lambda_0(\mathbf{K}_\infty^{(L)})n^{-1}$ *for some small constant* $c$ *and* $c_x = 2\sqrt{c_\sigma}Hc_{\mu,0}$ *then with probability at least* $1 - \delta$, *we have:*

$$\left\|\mathbf{\Theta}^{(L)}(t) - \mathbf{\Theta}^{(L)}(0)\right\|_2 \leq \frac{\lambda_0(\mathbf{K}_\infty^{(L)})}{4}.$$

**Lemma A.6.** *If* $R_\mathcal{S}(t') \leq \exp(-\lambda_0(\mathbf{K}_\infty^{(L)})t')R_\mathcal{S}(0)$ *holds for* $0 \leq t' \leq t$, *we have for any* $0 \leq t' \leq t$,

$$\left\|\mathbf{W}_\mu^{(l)}(t') - \mathbf{W}_\mu^{(l)}(0)\right\|_F \leq R'\sqrt{m},$$

*where* $R' = \frac{16c_{x,0}(c_x)^L\sqrt{n}\|\mathbf{y}-\hat{f}(\mathbf{X},0)\|_2}{\lambda_0(\mathbf{K}_\infty^{(L)})\sqrt{m}}$, *for some small constant* $c$ *with* $c_x = \max\{2\sqrt{c_\sigma}Lc_{\mu,0}, 1\}$.

*Proof of Lemma A.4.* The proof sketch is by induction method.

For $l = 0$, where the target is input which is fixed, thus satisfying the hypothesis. Now suppose the induction hypothesis holds for $l' = 0, \dots, l - 1$, we consider $l' = l$.

$$
\begin{aligned}
&\left\|\hat{\mathbf{x}}^{(l)}(t) - \hat{\mathbf{x}}^{(l)}(0)\right\|_2 \\
=&\frac{1}{\sqrt{m}}\left\|\mathbb{E}_{\boldsymbol{\xi}^{(l)}}\sigma(\mathbf{W}^{(l)}(t)\hat{\mathbf{x}}^{(l-1)}(t)) - \mathbb{E}_{\boldsymbol{\xi}^{(l)}}\sigma(\mathbf{W}^{(l)}(0)\hat{\mathbf{x}}^{(l-1)}(0))\right\|_2 \\
\leq&\frac{1}{\sqrt{m}}\left\|\mathbb{E}_{\boldsymbol{\xi}^{(l)}}\sigma(\mathbf{W}^{(l)}(t)\hat{\mathbf{x}}^{(l-1)}(t)) - \mathbb{E}_{\boldsymbol{\xi}^{(l)}}\sigma(\mathbf{W}^{(l)}(t)\hat{\mathbf{x}}^{(l-1)}(0))\right\|_2 \\
&+ \frac{1}{\sqrt{m}}\left\|\mathbb{E}_{\boldsymbol{\xi}^{(l)}}\sigma(\mathbf{W}^{(l)}(t)\hat{\mathbf{x}}^{(l-1)}(0)) - \mathbb{E}_{\boldsymbol{\xi}^{(l)}}\sigma(\mathbf{W}^{(l)}(0)\hat{\mathbf{x}}^{(l-1)}(0))\right\|_2 \\
\leq&\frac{1}{\sqrt{m}}H\left(\left\|\mathbb{E}_{\boldsymbol{\xi}^{(l)}}\mathbf{W}^{(l)}(0)\right\|_2 + \left\|\mathbb{E}_{\boldsymbol{\xi}^{(l)}}\mathbf{W}^{(l)}(t) - \mathbb{E}_{\boldsymbol{\xi}^{(l)}}\mathbf{W}^{(l)}(0)\right\|_F\right)\cdot\left\|\hat{\mathbf{x}}^{(l-1)}(t) - \hat{\mathbf{x}}^{(l-1)}(0)\right\|_2 \\
&+ \frac{1}{\sqrt{m}}H\left(\left\|\mathbb{E}_{\boldsymbol{\xi}^{(l)}}\mathbf{W}^{(l)}(t) - \mathbb{E}_{\boldsymbol{\xi}^{(l)}}\mathbf{W}^{(l)}(0)\right\|_F\right)\|\hat{\mathbf{x}}^{h-1}(0)\|_2 \\
\leq&\frac{1}{\sqrt{m}}H\left(c_{\mu,0}\sqrt{m} + R\sqrt{m}\right)HRc_{x,0}g_{c_x}(l-1) + \frac{1}{\sqrt{m}}H\sqrt{m}Rc_{x,0} \\
\leq&HRc_{x,0}\left(c_xg_{c_x}(l-1) + 1\right) \\
\leq&HRc_{x,0}g_{c_x}(l).
\end{aligned}
$$

$\square$

*Proof of Lemma A.5.* For simplicity, we define $z_{i,r}(t) = \mathbf{w}_r^{(L-1)}(t)^\top\hat{\mathbf{x}}_i^{(L-2)}(t)$.

Now we bound the distance between $\boldsymbol{\Theta}_{\mu,ij}^{(L)}(t)$ and $\boldsymbol{\Theta}_{\mu,ij}^{(L)}(0)$ through the following inequality:

$$
\left| \boldsymbol{\Theta}_{\mu,ij}^{(L)}(t) - \boldsymbol{\Theta}_{\mu,ij}^{(L)}(0) \right|
$$

$$
= \left| \frac{1}{m} \sum_{r=1}^{m} \mathbb{E}_{\boldsymbol{\xi}^{(l-1)}} \sigma\left(z_{i,r}(t)\right) \mathbb{E}_{\boldsymbol{\xi}^{(l-1)}} \sigma\left(z_{j,r}(t)\right) - \frac{1}{m} \sum_{r=1}^{m} \mathbb{E}_{\boldsymbol{\xi}^{(l-1)}} \sigma\left(z_{i,r}(0)\right) \mathbb{E}_{\boldsymbol{\xi}^{(l-1)}} \sigma\left(z_{j,r}(0)\right) \right|
$$

$$
\leq \frac{1}{m} \sum_{r=1}^{m} \left| \mathbb{E}_{\boldsymbol{\xi}^{(l-1)}} \sigma\left(z_{i,r}(t)\right) \mathbb{E}_{\boldsymbol{\xi}^{(l-1)}} \sigma\left(z_{j,r}(t)\right) - \mathbb{E}_{\boldsymbol{\xi}^{(l-1)}} \sigma\left(z_{i,r}(t)\right) \mathbb{E}_{\boldsymbol{\xi}^{(l-1)}} \sigma\left(z_{j,r}(0)\right) \right|
$$

$$
+ \frac{1}{m} \sum_{r=1}^{m} \left| \mathbb{E}_{\boldsymbol{\xi}^{(l-1)}} \sigma\left(z_{i,r}(t)\right) \mathbb{E}_{\boldsymbol{\xi}^{(l-1)}} \sigma\left(z_{j,r}(0)\right) - \mathbb{E}_{\boldsymbol{\xi}^{(l-1)}} \sigma\left(z_{i,r}(0)\right) \mathbb{E}_{\boldsymbol{\xi}^{(l-1)}} \sigma\left(z_{j,r}(0)\right) \right|
$$

$$
\leq \frac{\beta H}{m} \left( \sum_{r=1}^{m} \mathbb{E}_{\boldsymbol{\xi}^{(l-1)}} \left| z_{i,r}(t) - z_{i,r}(0) \right| + \mathbb{E}_{\boldsymbol{\xi}^{(l-1)}} \left| z_{j,r}(t) - z_{j,r}(0) \right| \right)
$$

$$
\leq \frac{\beta H}{\sqrt{m}} \sqrt{ \sum_{r=1}^{m} \left| \mathbb{E}_{\boldsymbol{\xi}^{(l-1)}} z_{i,r}(t) - \mathbb{E}_{\boldsymbol{\xi}^{(l-1)}} z_{i,r}(0) \right|^2 }
$$

$$
\leq n \beta H c_{x,0} g_{c_x}(L) R.
$$

where we have used the result in Lemma A.4. Therefore we can bound the perturbation:

$$
\left\| \boldsymbol{\Theta}_{\mu}^{(L)}(t) - \boldsymbol{\Theta}_{\mu}^{(L)}(0) \right\|_F = \sqrt{ \sum_{i,j=1}^{n} \left| \boldsymbol{\Theta}_{\mu,ij}^{(L)}(t) - \boldsymbol{\Theta}_{\mu,ij}^{(L)}(0) \right|^2 } \leq \beta H c_{x,0} g_{c_x}(L) R.
$$

Recall the bound on $R$, which is $R \leq c g_{c_x}(L)^{-1} \lambda_0(\mathbf{K}_\infty^{(L)}) n^{-1}$, we have the desired result:

$$
\left\| \boldsymbol{\Theta}^{(L)}(t) - \boldsymbol{\Theta}^{(L)}(0) \right\|_2 \leq \frac{\lambda_0(\mathbf{K}_\infty^{(L)})}{4}.
$$

$\square$

*Proof of Lemma A.6.* We first consider the derivative of $\mathbf{W}_{\mu}^{(l)}$ and have:

$$
\left\| \frac{d}{ds} \mathbf{W}_{\mu}^{(l)}(s) \right\|_F
$$

$$
= \eta \left\| \left( \frac{1}{m} \right)^{\frac{L-l+1}{2}} \sum_{i=1}^{n} (y_i - \hat{f}(\mathbf{x}_i; s)) \hat{\mathbf{x}}_i^{(l-1)}(s) \mathbb{E}_{\boldsymbol{\xi}^{(k)}} \left( \prod_{k=l+1}^{L} \mathbf{J}_i^{(k)}(s) \mathbf{W}^{(k)}(s) \right) \mathbf{J}_i^{(l)}(s) \right\|_F
$$

$$
\leq \eta \left( \frac{1}{m} \right)^{\frac{L-l+1}{2}} \sum_{i=1}^{n} \left| y_i - \hat{f}(\mathbf{x}_i; s) \right| \left\| \hat{\mathbf{x}}_i^{(l-1)}(s) \right\|_2 \prod_{k=l+1}^{L} \left\| \mathbb{E}_{\boldsymbol{\xi}^{(k)}} \mathbf{W}^{(k)}(s) \right\|_2 H^{L-l+1},
$$

where

$$
\mathbf{J}^{(l')} \triangleq \mathrm{diag}\left( \sigma'\left( (\mathbf{w}_1^{(l')})^\top \mathbf{x}^{(l'-1)} \right), \ldots, \sigma'\left( (\mathbf{w}_m^{(l')})^\top \mathbf{x}^{(l'-1)} \right) \right) \in \mathbb{R}^{m \times m}
$$

are the derivative matrices induced by the activation function and we have used $\left\| \mathbf{J}^{(k)}(s) \right\|_2 \leq H$.

To bound $\left\| \hat{\mathbf{x}}_i^{(l-1)}(s) \right\|_2$, by Lemma A.4 we get:

$$
\left\| \hat{\mathbf{x}}_i^{(l-1)}(s) \right\|_2 \leq H c_{x,0} g_{c_x}(l) R + c_{x,0} \leq 2 c_{x,0}.
$$

To bound $\left\|\mathbb{E}_{\boldsymbol{\xi}^{(k)}}\mathbf{W}^{(k)}(s)\right\|_2 = \left\|\mathbf{W}_\mu^{(k)}(s)\right\|_2$, we have:

$$
\prod_{k=l+1}^{L}\left\|\mathbf{W}_\mu^{(k)}(s)\right\|_2 \leq \prod_{k=l+1}^{L}\left(\left\|\mathbf{W}_\mu^{(k)}(0)\right\|_2 + \left\|\mathbf{W}_\mu^{(k)}(s) - \mathbf{W}_\mu^{(k)}(0)\right\|_2\right)
$$

$$
\leq \prod_{k=l+1}^{L}(c_{\mu,0}\sqrt{m} + R'\sqrt{m})
$$

$$
= (c_{\mu,0} + R')^{L-l}\, m^{\frac{L-l}{2}}
$$

$$
\leq (2c_{\mu,0})^{L-l}\, m^{\frac{L-l}{2}}.
$$

Plugging in these two bounds back, we obtain

$$
\left\|\frac{d}{ds}\mathbf{W}_\mu^{(l)}(s)\right\|_F \leq 4\eta c_{x,0}c_x^L \sum_{i=1}^{n}|y_i - \hat{f}(\mathbf{x}_i; s)|
$$

$$
\leq 4\eta c_{x,0}c_x^L\sqrt{n}\left\|\mathbf{y} - \hat{f}(\mathbf{X}; s)\right\|_2
$$

$$
\leq e^{-\lambda_0 s}\eta\lambda_0(\mathbf{K}_\infty^{(L)})R'\sqrt{m}.
$$

Integrating the derivative of weights, we obtain:

$$
\left\|\mathbf{W}_\mu^{(l)}(s) - \mathbf{W}_\mu^{(l)}(0)\right\|_F \leq \int_{s'=0}^{s}\left\|\frac{d}{ds'}\mathbf{W}_\mu^{(l)}(s')\right\|_F \leq R'\sqrt{m}.
$$

$\square$

### A.3   Setp 3. Towards linear convergence rate of expected empirical loss

Now we process to analyze the convergence rate of expected empirical error. Combined with fact that least eigenvalue of PNTKs and change of weights are bounded during training, the behavior of the loss is traceable. To finalize the proof for Theorem 4.2, we show:

**Lemma A.7.** *If $R' < R$, we have $\underline{R}_{\mathcal{S}}(t) \leq \exp(-\lambda_0(\mathbf{K}_\infty^{(L)})t)\underline{R}_{\mathcal{S}}(0)$.*

*Proof of Lemma A.7.* According to the gradient flow of output function, we have:

$$
\frac{d\hat{f}(\mathbf{x}_i, t)}{dt} = \sum_{l=1}^{L}\left(\left\langle\frac{\partial\hat{f}(\mathbf{x}_i; t)}{\partial\mathbf{W}_\mu^{(l)}}, \frac{d\mathbf{W}_\mu^{(l)}(t)}{dt}\right\rangle + \left\langle\frac{\partial\hat{f}(\mathbf{x}; t)}{\partial\mathbf{W}_\sigma^{(l)}}, \frac{d\mathbf{W}_\sigma^{(l)}(t)}{dt}\right\rangle\right)
$$

$$
= \sum_{j=1}^{n}(\mathbf{y}_i - \hat{f}(\mathbf{x}_j))\sum_{l=1}^{L}\left(\left\langle\frac{\partial\hat{f}(\mathbf{x}_i)}{\partial\mathbf{W}_\mu^{(l)}}, \frac{\hat{f}(\mathbf{x}_j)}{\partial\mathbf{W}_\mu^{(l)}}\right\rangle + \left\langle\frac{\partial\hat{f}(\mathbf{x}_i)}{\partial\mathbf{W}_\sigma^{(l)}}, \frac{\partial\hat{f}(\mathbf{x}_j)}{\partial\mathbf{W}_\sigma^{(l)}}\right\rangle\right)
$$

$$
= \sum_{j=1}^{n}(\mathbf{y}_j - \hat{f}(\mathbf{x}_j; t))(\boldsymbol{\Theta}_{\mu,ij} + \boldsymbol{\Theta}_{\sigma,ij}).
$$

Then the dynamics of loss can be calculated:

$$
\frac{d}{dt}R_{\mathcal{S}}(t) = \frac{1}{2}\frac{d}{dt}\left\|\hat{f}(\mathbf{X}; t) - \mathbf{y}\right\|_2^2
$$

$$
= -\left(\mathbf{y} - \hat{f}(\mathbf{X}, t)\right)^\top(\boldsymbol{\Theta}_\mu(t) + \boldsymbol{\Theta}_\sigma(t))\left(\mathbf{y} - \hat{f}(\mathbf{X}, t)\right)
$$

$$
\leq -\lambda_0\left(\boldsymbol{\Theta}_\mu(t) + \boldsymbol{\Theta}_\sigma(t)\right)\left\|\mathbf{y} - \hat{f}(\mathbf{X}; t)\right\|_2^2
$$

$$
\leq -\lambda_0\left(\boldsymbol{\Theta}_\mu^{(L)}(t) + \boldsymbol{\Theta}_\sigma^{(L)}(t)\right)\left\|\mathbf{y} - \hat{f}(\mathbf{X}; t)\right\|_2^2
$$

$$
\leq -\lambda_0\left(\mathbf{K}_\infty^{(L)}\right)\left\|\mathbf{y} - \hat{f}(\mathbf{X}; t)\right\|_2^2,
$$

where we have used the condition $R' < R$ and $\lambda_0(\mathbf{\Theta}_\mu^{(L)}(t) + \mathbf{\Theta}_\sigma^{(L)}(t)) \leq \lambda_0(\mathbf{\Theta}_\mu(t) + \mathbf{\Theta}_\sigma(t))$. Therefore, we have the desired result:

$$\underline{R}_\mathcal{S}(t) \leq \exp(-\lambda_0(\mathbf{K}_\infty^{(L)})t)\underline{R}_\mathcal{S}(0).$$

Finally, we provide a bound for $\widehat{R}_S(Q(0))$:

$$\left\|\mathbf{y} - \hat{f}(\mathbf{X}, 0)\right\|_2^2 = \sum_{i=1}^n \left(y_i^2 + y_i\hat{f}(\mathbf{x}_i, 0) + f(\mathbf{x}_i, 0)^2\right) = \sum_{i=1}^n(1 + O(1)) = O(n).$$

Recall that in Lemma A.5 and Lemma A.6:

$$R \leq cg_{c_x}(L)^{-1}\lambda_0(\mathbf{K}_\infty^{(L)})n^{-1}, \quad R' = \frac{16c_{x,0}\,(c_x)^L\,\sqrt{n}\|\mathbf{y} - \hat{f}(\mathbf{X}, 0)\|_2}{\lambda_0(\mathbf{K}_\infty^{(L)})\sqrt{m}}.$$

Thus $R' < R$ yields $m = \Omega\left(\frac{n^4 2^{O(L)}}{\lambda_0^4(\mathbf{K}_\infty^{(L)})}\right)$.

$\square$

# B    Proof of Theorem 4.3

**Theorem B.1** (Restatement of Theorem 4.3). *Suppose $m \geq \text{poly}(n, 1/\lambda_0, 1/\delta, 1/\mathcal{E})$ and the objective function follows the form (17), for any test input $\mathbf{x}_{te} \in \mathbb{R}^d$ with probability at least $1 - \delta$ over the random initialization, we have*

$$\hat{f}(\mathbf{x}_{te}; t)|_{t=\infty} = \mathbf{\Theta}_\mu^\infty(\mathbf{x}, \mathbf{X})(\mathbf{\Theta}_\mu^\infty(\mathbf{X}, \mathbf{X}) + \lambda/c_\sigma^2\mathbf{I})^{-1}\mathbf{y} \pm \mathcal{E}, \tag{25}$$

*where $\mathcal{E}$ is a small error term between output function of finite-width PNN and infinite-width PNN.*

*Proof of Theorem B.1.* To proceed the proof, we first establish the result of kernel ridge regression in the infinite-width limit, and then bound the perturbation on the predict. According the linearization rules for infinitely-wide networks (Lee et al., 2019), the output function can be expressed as,

$$\hat{f}^\infty(\mathbf{x}, t) = \boldsymbol{\phi}_\mu(\mathbf{x})^\top(\boldsymbol{\theta}_\mu(t) - \boldsymbol{\theta}_\mu(0)) + \boldsymbol{\phi}_\sigma(\mathbf{x})^\top(\boldsymbol{\theta}_\sigma(t) - \boldsymbol{\theta}_\sigma(0)),$$

where $\boldsymbol{\phi}_\mu(\mathbf{x}) = \nabla_{\boldsymbol{\theta}_\mu}\hat{f}(\mathbf{x}; 0)$, and $\boldsymbol{\phi}_\sigma(\mathbf{x}) = \nabla_{\boldsymbol{\theta}_\sigma}\hat{f}(\mathbf{x}; 0)$. Recall that we assume that $\boldsymbol{\theta}_\sigma$ does not change during training, then the KL divergence reduces to

$$\text{KL} = \frac{1}{2nc_\sigma^2}\|\boldsymbol{\theta}_\mu(t) - \boldsymbol{\theta}_\mu(0)\|_2^2.$$

Then the gradient flow equation for $\boldsymbol{\theta}_\mu$ becomes,

$$\frac{d\boldsymbol{\theta}_\mu(t)}{dt} = \frac{\partial L(t)}{\partial\boldsymbol{\theta}_\mu(t)} = \left(\mathbf{\Theta}_\mu^\infty + \lambda/c_\sigma^2\mathbf{I}\right)\left(\boldsymbol{\theta}^{(\mu)}(t) - \boldsymbol{\theta}^{(\mu)}(0)\right) + \boldsymbol{\phi}_\mu(\mathbf{X})\left(\hat{f}^\infty(\mathbf{X}; 0) - \mathbf{y}\right),$$

which is an ordinary differential equation, and the solution is,

$$\boldsymbol{\theta}_\mu(t) = \boldsymbol{\phi}_\mu(\mathbf{X})^\top\left(\mathbf{\Theta}_\mu^\infty(\mathbf{X}, \mathbf{X}) + \lambda/c_\sigma^2\mathbf{I}\right)^{-1}\left(\mathbf{I} - e^{-(\mathbf{\Theta}_\mu^\infty(\mathbf{X},\mathbf{X})+\lambda/c_\sigma^2\mathbf{I})t}\right)\mathbf{y}.$$

Plug this result into the linearization of expected output function, we have,

$$\hat{f}^\infty(\mathbf{x}; t) = \mathbf{\Theta}_\mu^\infty(\mathbf{x}, \mathbf{X})(\mathbf{\Theta}_\mu^\infty(\mathbf{X}, \mathbf{X}) + \lambda/c_\sigma^2\mathbf{I})^{-1}(\mathbf{I} - e^{-(\mathbf{\Theta}_\mu^\infty(\mathbf{X},\mathbf{X})+\lambda/c_\sigma^2\mathbf{I})t})\mathbf{y}.$$

Then we take the time to be infinity and have:

$$\hat{f}^\infty(\mathbf{x})|_{t=\infty} = \mathbf{\Theta}_\mu^\infty(\mathbf{x}, \mathbf{X})(\mathbf{\Theta}_\mu^\infty(\mathbf{X}, \mathbf{X}) + \lambda/c_\sigma^2\mathbf{I})^{-1}\mathbf{y}.$$

The next step is to show the difference between finite-width neural network and infinitely-wide network:

$$\left| \hat{f}(\mathbf{x}_{te}) - \hat{f}^{\infty}(\mathbf{x}_{te}) \right| \le O(\mathcal{E}),$$

where $\mathcal{E} = \mathcal{E}_{\text{init}} + \frac{\sqrt{n}\mathcal{E}_{\Theta}}{\lambda_0 + \beta}$ with $\left\| \hat{f}(\mathbf{x}_{te}; 0) \right\|_2 \le \mathcal{E}_{\text{init}}$ and $\|\mathbf{\Theta}_{\mu}^{\infty} - \mathbf{\Theta}_{\mu}(t)\|_2 \le \mathcal{E}_{\Theta}$. Note the expression of output function in the infinite-width limit can be rewritten as $\hat{f}^{\infty}(\mathbf{x}_{te}) = \phi(\mathbf{x}_{te})^{\top}\boldsymbol{\beta}$ and the solution to this equation can be further written as the result of applying gradient flow on the following kernel ridge regression problem

$$\min_{\boldsymbol{\beta}} \sum_{i=1}^{n} \frac{1}{2n} \left\| \phi(\mathbf{x}_i)^{\top}\boldsymbol{\beta} - \mathbf{x}_i \right\|_2^2 + \lambda/c_{\sigma}^2 \|\boldsymbol{\beta}\|_2^2,$$

with initialization $\boldsymbol{\beta}(0) = 0$. We use $\boldsymbol{\beta}(t)$ to denote this parameter at time $t$ trained by gradient flow and $\hat{f}^{\infty}(\mathbf{x}_{te}, \boldsymbol{\beta}(t))$ be the predictor for $\mathbf{x}_{te}$ at time $t$. With these notations, we rewrite

$$\hat{f}^{\infty}(\mathbf{x}_{te}) = \int_{t=0}^{\infty} \frac{d\hat{f}(\boldsymbol{\beta}(t), \mathbf{x}_{te})}{dt} dt,$$

where we have used the fact that the initial prediction is 0.

We thus can analyze the difference between the PNN predictor and infinite-width PNN predictor via this integral form as follows:

$$\left\| \hat{f}^{\infty}(\mathbf{x}_{te}) - \hat{f}(\mathbf{x}_{te}) \right\|_2$$

$$\le \left\| \hat{f}(\boldsymbol{\theta}_{\mu}(0), \mathbf{x}_{te}) \right\|_2 + \left\| \int_{t=0}^{\infty} \left( \frac{d\hat{f}(\boldsymbol{\theta}_{\mu}(t), \mathbf{x}_{te})}{dt} - \frac{d\hat{f}^{\infty}(\boldsymbol{\beta}(t), \mathbf{x}_{te})}{dt} \right) dt \right\|_2$$

$$= \left\| \hat{f}(\boldsymbol{\theta}(0), \mathbf{x}_{te}) \right\|_2 + \left\| -\frac{1}{n} \int_{t=0}^{\infty} (\mathbf{\Theta}_{\mu}^{\infty}(\mathbf{x}_{te}, \mathbf{X})^{\top} (\hat{f}^{\infty}(t) - \mathbf{y}) - \mathbf{\Theta}_{\mu}(\mathbf{x}_{te}, \mathbf{X}; t)^{\top} (\hat{f}(t) - \mathbf{X})) dt \right.$$

$$\left. - \lambda/c_{\sigma}^2 \int_{t=0}^{\infty} (\phi^{\infty}(\mathbf{x}_{te})^{\top}\boldsymbol{\beta}(t) - \phi_{\mu}(\mathbf{x}_{te}; t)^{\top}\overline{\boldsymbol{\theta}}(t) dt \right\|_2$$

$$\le \mathcal{E}_{init} + \left\| \frac{1}{n} \int_{t=0}^{\infty} (\mathbf{\Theta}_{\mu}(\mathbf{x}_{te}, \mathbf{X}; t) - \mathbf{\Theta}^{\infty}(\mathbf{x}_{te}, \mathbf{X}))^{\top} (\hat{f}(t) - \mathbf{X}) dt + \beta \int_{t=0}^{\infty} (\phi_{\mu}(\mathbf{x}_{te}, t) - \phi^{\infty}(\mathbf{x}_{te}))^{\top} \boldsymbol{\beta}(t) dt \right\|_2$$

$$+ \left\| \frac{1}{n} \int_{t=0}^{\infty} \mathbf{\Theta}_{\mu}^{\infty}(\mathbf{x}_{te}, \mathbf{X})^{\top} (\hat{f}^{\infty}(t) - \hat{f}(t)) dt + \lambda/c_{\sigma}^2 \int_{t=0}^{\infty} (\phi_{\mu}^{\infty}(\mathbf{x}_{te}))^{\top} (\boldsymbol{\beta}(t) - \overline{\boldsymbol{\theta}}_{\mu}(t)) dt \right\|_2$$

$$\le \mathcal{E}_{init} + \left( \max_{0 \le t \le \infty} \|\mathbf{\Theta}_{\mu}(\mathbf{x}_{te}, \mathbf{X}; t) - \mathbf{\Theta}_{\mu}^{\infty}(\mathbf{x}_{te}, \mathbf{X})\|_2 \int_{t=0}^{\infty} \left\| \hat{f}(t) - \mathbf{y} \right\|_2 dt \right.$$

$$+ \lambda/c_{\sigma}^2 \max_{0 \le t \le \infty} \|\phi_{\mu}(\mathbf{x}_{te}; t) - \phi_{\mu}^{\infty}(\mathbf{x}_{te})\|_2 \int_{t=0}^{\infty} \|\boldsymbol{\beta}\|_2 dt \right) + \left( \max_{0 \le t \le \infty} \|\mathbf{\Theta}_{\mu}^{\infty}(\mathbf{x}_{te}, \mathbf{X})\|_2 \int_{t=0}^{\infty} \|\hat{f}(t) - \hat{f}^{\infty}(t)\|_2 dt \right.$$

$$\left. + \lambda/c_{\sigma}^2 \max_{0 \le t \le \infty} \|\phi_{\mu}^{\infty}(\mathbf{x}_{te})\|_2 \int_{t=0}^{\infty} \|\boldsymbol{\beta}(t) - \overline{\boldsymbol{\theta}}_{\mu}(t)\|_2 dt \right)$$

$$\triangleq \mathcal{E}_{init} + I_2 + I_3,$$

where $\overline{\boldsymbol{\theta}}_{\mu}(t) = \boldsymbol{\theta}_{\mu}(t) - \boldsymbol{\theta}_{\mu}(0)$. For the second term $I_2$, we know that $\|\hat{f}(t) - \mathbf{y}\|_2^2 + \beta\|\overline{\boldsymbol{\theta}}_{\mu}\|_2^2 \le \exp(-(\lambda_0 + \lambda/c_{\sigma}^2)t)\|\hat{f}(0) - \mathbf{y}\|_2^2$.

Therefore, we can bound:

$$\int_0^{\infty} \|\hat{f}(t) - \mathbf{y}\|_2 + \beta\|\overline{\boldsymbol{\theta}}_{\mu}(t)\|_2 dt$$

$$\le \int_{t=0}^{\infty} \exp(-(\lambda_0 + \lambda/c_{\sigma}^2)t)(\|\hat{f}(0) - \mathbf{y}\|_2) dt$$

$$= O\left( \frac{\sqrt{n}}{\lambda_0 + \lambda/c_{\sigma}^2} \right).$$

As a result, we have $I_2 = O\left(\frac{\sqrt{n}\mathcal{E}_\Theta}{\lambda_0 + \lambda/c_\sigma^2}\right)$. To bound $I_3$, we have

$$\int_0^\infty \|\hat{f}(t) - \hat{f}^\infty(t)\|_2 + \lambda/c_\sigma^2\|\boldsymbol{\beta} - \overline{\boldsymbol{\theta}}_\mu\|_2 dt$$

$$\leq \int_0^\infty \|\hat{f}(t) - \mathbf{y}\|_2 + \lambda/c_\sigma^2\|\overline{\boldsymbol{\theta}}_\mu\|_2 dt + \int_0^\infty \|\hat{f}^\infty(t) - \mathbf{X}\|_2 + \lambda/c_\sigma^2\|\boldsymbol{\beta}\|_2 dt$$

$$\leq O\left(\frac{\sqrt{n}}{\lambda_0 + \lambda/c_\sigma^2}\right).$$

As a result, we have $I_3 = O\left(\frac{\sqrt{n}\mathcal{E}_\Theta}{\lambda_0 + \lambda/c_\sigma^2}\right)$. Lastly, we put things together and get

$$|\hat{f}(t) - \hat{f}^\infty(t)| \leq O\left(\mathcal{E}_{init} + \mathcal{E}_\Theta \frac{\sqrt{n}}{\lambda_0 + \beta}\right).$$

$\square$

## B.1 Proof of Theorem 4.4

*Proof of Theorem 4.4.* Our proof is based on a characterization of the margin $\mathcal{M}_Q^\mathcal{S}$ and $\mathcal{M}_{Q^2}^\mathcal{S}$ via the explicit solution found in Theorem 4.3:

$$\mathcal{M}_Q^\mathcal{S} = \frac{1}{n}\sum_{i=1}^n y_i \cdot \mathbb{E}_{h\sim Q}h(\mathbf{x}_i) = \frac{1}{n}\sum_{i=1}^n y_i\hat{f}^\infty(\mathbf{x}_i),$$

$$\mathcal{M}_{Q^2}^S = \frac{1}{n}\sum_{i=1}^n \mathbb{E}_{h,h'\sim Q^2}h(\mathbf{x}_i)h'(\mathbf{x}_i) = \frac{1}{n}\sum_{i=1}^n\sum_{j=1}^n \hat{f}^\infty(\mathbf{x}_i)\hat{f}^\infty(\mathbf{x}_j).$$

Then by Theorem 3.3, we have:

$$R_\mathcal{D}(B_Q) \leq 1 - \frac{\left(\frac{1}{n}\sum_{i=1}^n y_i\hat{f}^\infty(\mathbf{x}_i) - 2B\frac{\sqrt{\ln(\frac{2n}{\delta})}}{\sqrt{2n}}\right)^2}{\frac{1}{n}\sum_{i=1}^n\sum_{j=1}^n \hat{f}^\infty(\mathbf{x}_i)\hat{f}^\infty(\mathbf{x}_j) + 2B^2\frac{\sqrt{\ln(\frac{2n}{\delta})}}{\sqrt{2n}}}.$$

$\square$

# C Appendix for Experiments

This section contains additional experimental results and derivation process for proxy. Training is performed with a server with a CPU with 5,120 cores, and a 32 GB Nvidia Quadro V100.

## C.1 Derivation of proxy

Our derivation is based on a characterization of the empirical error and KL divergence term via the explicit solution found in Theorem 4.3. The objective function consists of two terms, one is the empirical error, and another is KL divergence. (i) We first bound the empirical error $\sqrt{\sum_{i=1}^n(\hat{f}^\infty(\mathbf{x}_i, t = \infty) - y_i)^2}$ with following inequality,

$$\sqrt{\sum_{i=1}^n(\hat{f}^\infty(\mathbf{x}_i, t = \infty) - y_i)^2} = \left\|\boldsymbol{\Theta}_\mu^\infty(\mathbf{X}, \mathbf{X})(\boldsymbol{\Theta}_\mu^\infty(\mathbf{X}, \mathbf{X}) + \lambda/c_\sigma^2\mathbf{I})^{-1}\mathbf{y} - \mathbf{y}\right\|_2$$

$$= \left\|\lambda/c_\sigma^2\left(\boldsymbol{\Theta}_\mu^\infty(\mathbf{X}, \mathbf{X}) + \lambda/c_\sigma^2\mathbf{I}\right)^{-1}\mathbf{y}\right\|_2$$

$$= \lambda/c_\sigma^2\sqrt{\mathbf{y}^\top\left(\boldsymbol{\Theta}_\mu^\infty(\mathbf{X}, \mathbf{X}) + \lambda/\sigma_0^2\mathbf{I}\right)^{-2}\mathbf{y}}.$$

Then we bound the error term as follows:

$$
\begin{aligned}
\frac{1}{n}\sum_{i=1}^{n}\ell\big(\hat{f}^{\infty}(\mathbf{x}_i),y_i\big) &= \frac{1}{n}\sum_{i=1}^{n}\left[\ell(\hat{f}^{\infty}(\mathbf{x}_i),y_i)-\ell(y_i,y_i)\right]\\
&\leq \frac{1}{n}\sum_{i=1}^{n}\left|\hat{f}^{\infty}(\mathbf{x}_i)-y_i\right|\\
&\leq \frac{1}{\sqrt{n}}\sqrt{\sum_{i=1}^{n}\left|\hat{f}^{\infty}(\mathbf{x}_i)-y_i\right|^2}\\
&\leq \frac{\lambda}{c_\sigma^2}\sqrt{\frac{\mathbf{y}^{\top}(\boldsymbol{\Theta}_\mu^\infty(\mathbf{X},\mathbf{X})+\lambda/c_\sigma^2\mathbf{I})^{-2}\mathbf{y}}{n}}.
\end{aligned}
$$

(ii) The next step is to calculate the KL divergence. According to the solution of differential equation in Theorem B.1, we have:

$$
\boldsymbol{\theta}_\mu(t)|_{t=\infty}-\boldsymbol{\theta}_\mu(0)=\boldsymbol{\phi}_\mu(\mathbf{x})^{\top}\big(\boldsymbol{\Theta}_\mu^\infty(\mathbf{X},\mathbf{X})+\lambda/c_\sigma^2\mathbf{I}\big)^{-1}\mathbf{y}
$$

Therefore, the KL divergence is,

$$
\begin{aligned}
\mathrm{KL} &= 1/c_\sigma^2\cdot\mathbf{y}^{\top}\big(\boldsymbol{\Theta}_\mu^\infty(\mathbf{X},\mathbf{X})+\lambda/c_\sigma^2\mathbf{I}\big)^{-1}\boldsymbol{\Theta}_\mu^\infty(\mathbf{X},\mathbf{X})\big(\boldsymbol{\Theta}_\mu^\infty(\mathbf{X},\mathbf{X})+\lambda/c_\sigma^2\mathbf{I}\big)^{-1}\mathbf{y}\\
&\leq \frac{1}{c_\sigma^2}\mathbf{y}^{\top}\big(\boldsymbol{\Theta}_\mu^\infty(\mathbf{X},\mathbf{X})+\lambda/c_\sigma^2\mathbf{I}\big)^{-1}\mathbf{y}.
\end{aligned}
$$

Finally, by Equation 5, we have,

$$
\underline{R}_{\mathcal{S}}(Q)+\lambda\frac{\mathrm{KL}(\boldsymbol{\theta}(t)\|\boldsymbol{\theta}(0))}{n}\leq\frac{\mathbf{y}^{\top}\big(\boldsymbol{\Theta}_\mu^\infty(\mathbf{X},\mathbf{X})+\frac{\lambda}{c_\sigma^2}\mathbf{I}\big)^{-1}\mathbf{y}}{nc_\sigma^2}+\frac{\lambda}{c_\sigma^2}\sqrt{\frac{\mathbf{y}^{\top}\big(\boldsymbol{\Theta}_\mu^\infty(\mathbf{X},\mathbf{X})+\frac{\lambda}{c_\sigma^2}\mathbf{I}\big)^{-2}\mathbf{y}}{n}}.
$$

## C.2 Validation of theoretical results

We first provide empirical support showing that the training dynamics of wide probabilistic neural networks using the training objective derived from a PAC-Bayes bound are captured by PNTK, which validates Lemma A.6.

Consider a three hidden layer ReLU fully-connected network of the training objective derived from the PAC-Bayesian lambda bound in Equation (5), using an ordinary MSE function as loss. The neural network is trained with a full-batch gradient descent using learning rates equal to one on a fixed subset of MNIST ($|D|=128$) of ten classifications. A random initialized prior with no connection to data is used since it is in line with our theoretical setting and we only intend to observe the change in parameters rather than the performance of the actual bound.

After $T=2^{17}$ steps of gradient descent updates from different random initialization, we plot the changes of $\mathbf{W}_\mu^{(l)}$ and $\mathbf{W}_\sigma^{(l)}$ of input/output/hidden layer with respect to width $m$ for each layer on Figure 2. We observe that the relative Frobenius norm change in the input/output layer's weights scales as $1/\sqrt{m}$ while the hidden layers' weight scales is $1/m$ during the training, which verifies Lemma A.6.

## C.3 Comparison of gradient norm with respect to mean weight and variance weight

We then conduct an experiment to compare the gradient of norm with respect to $\boldsymbol{\theta}_\mu$ and $\boldsymbol{\theta}_\sigma$. The result is shown in Figure 3. We can see that the gradient norm of $\nabla_{\boldsymbol{\theta}_\mu}f(\mathbf{x})$ is much larger than that of $\nabla_{\boldsymbol{\theta}_\sigma}f(\mathbf{x})$, which implies that $\boldsymbol{\theta}_\sigma$ is effectively fixed during gradient descent training.

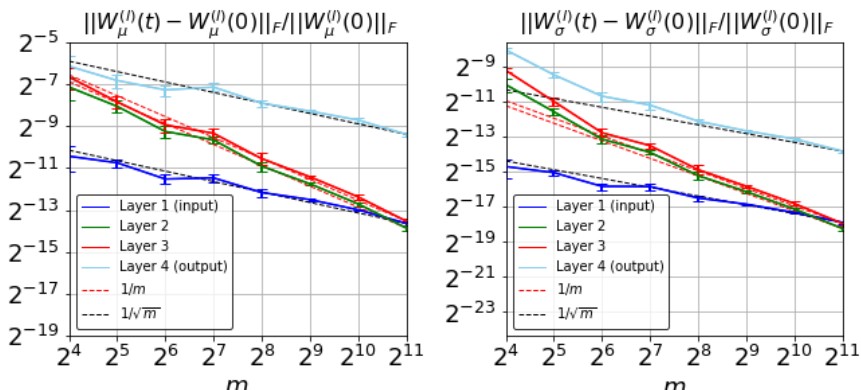

Figure 2: Relative Frobenius norm change in $\mu$ and $\sigma$ respectively during training with MSE loss which is derived from the classic PAC-Bayesian bound, where $m$ is the width of the network.

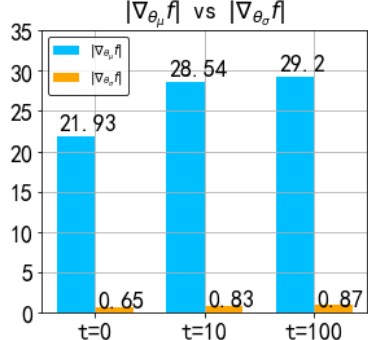

Figure 3: Comparison between the norm of derivative of output to mean and variance weights.

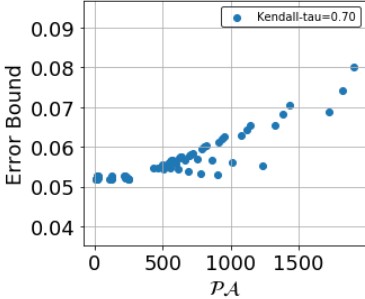

Figure 4: Correlation between aggregated proxy $\mathcal{PA}$ and generalization bound.

## C.4   Correlation between generalization bound proxy metric and generalization bound

In Figure 1, we observe a positive and significant correlation between $\mathcal{PA}$ and generalization bound held among different values of a selected hyperparameter while fixing other hyperparameters. Furthermore, we provide a Figure 4 presenting the correlation for aggregated values of $\rho_0$ and $\lambda$, under the circumstance where 50% data is used for prior training. We can clearly see that lower $\mathcal{PA}$ corresponds to the lower bound, with a strong positive Kendall-tau correlation of 0.7.

### C.5 Grid search

For selecting hyperparameters, we conduct a grid search over $\rho_0$, percent of prior data, and KL penalty $\lambda$. Notably, we do grid sweep over the data for prior training with different proportion in [0.2, 0.3, 0.4, 0.5, 0.6, 0.7, 0.8, 0.9] since 0.2 is the minimum proportion required for obtaining a reasonably lower value generalization bound (Dziugaite et al., 2021). For the rest, we run over $\rho_0$ at value [0.03, 0.05, 0.07, 0.09, 0.1, 0.3, 0.5, 0.7] for FCN ([0.05, 0.07, 0.09, 0.1, 0.3, 0.5, 0.7, 0.9] for CNN) and KL penalty at [0.0001, 0.0005, 0.001, 0.005, 0.01, 0.05, 0.1, 0.5, 1] for both structures.

### C.6 Scability of Proxy

In our proposed method for hyperparameter selection, we choose a subset of data for computing NTK to balance computational efficiency and accuracy. To investigate the scalability of $\mathcal{PA}$, we carry out additional experiments to explore the relationship between the error bound and $\mathcal{PA}$ concerning the number of samples per class. The experimental results, as depicted in Figure 5, demonstrate that when the size of the data subset surpasses a certain threshold (125 samples per class), the performance of $\mathcal{PA}$ with different hyperparameters stabilizes.

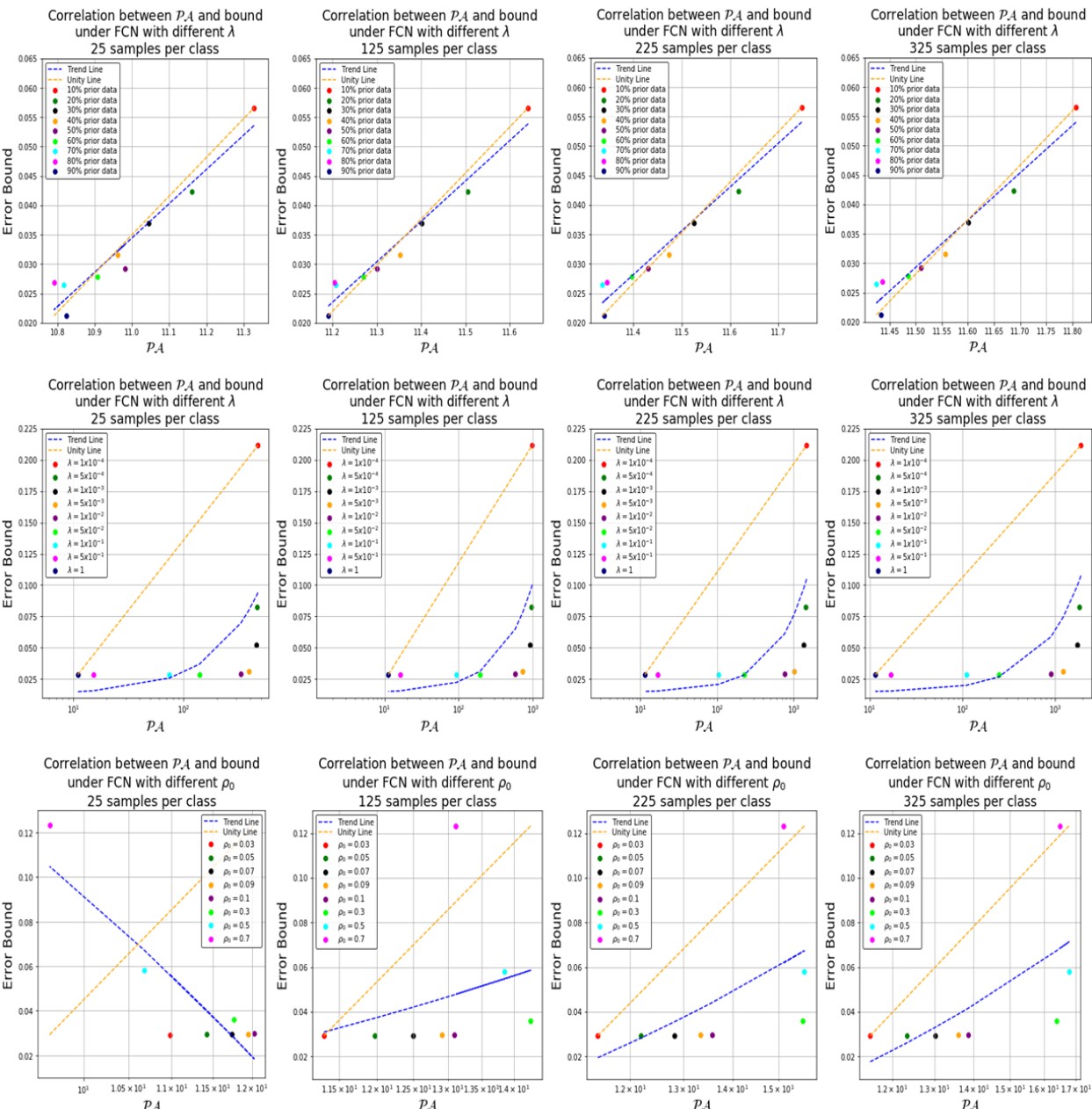

Figure 5: The correlation between $\mathcal{PA}$ and error bound concerning the data size.

