# OpenReview forum: "Analyzing Deep PAC-Bayesian Learning with Neural Tangent Kernel: Convergence, Analytic Generalization Bound, and Efficient Hyperparameter Selection"
_TMLR — Accepted by TMLR_

### Review · Reviewer_PSU9 · 2023-03-11

**Summary Of Contributions:**

The paper reports a PAC-Bayesian analysis for deep neural networks with randomized weights. Starting from the well-established generic PAC-Bayesian bounds, the paper develops a theoretical analysis of the convergence behavior, the characteristics of the resulting function approximation as width goes infinity, and the resulting generalization bound.

**Audience:**

Yes

**Claims And Evidence:**

Yes

**Requested Changes:**

There may be a typo on the r.h.s. of Eq (4). Should there really be a multiplicative empirical risk term there

While the PAC Bayes aspect is a clear novelty, generalization behavior of NTKs has been previously analyzed. The paper will be more complete if some of this body of work is cited and the advantages of the PAC-Bayesian framework on the state of the art there are clarified. For instance:

M. Nonnenmacher et al., Which Minimizer Does My Neural Network Converge To?, ECML-PKDD, 2021

---
After rebuttal: The author response addressed my concerns satisfactorily.

**Strengths And Weaknesses:**

 * This is to my knowledge the first attempt to analyze probabilistic neural networks using the PAC-Bayesian theory and from a Neural Tangent Kernel point of view. Hence the content is novel.

 * Providing generalization guarantees for deep neural networks is an important and interesting problem with significant potential to impact real-world applications as safety-critical use cases.

 * The paper exhibits an excellent scientific writing practice. It is easy and enjoyable to read and technically precise. It also gives insightful verbal descriptions of the presented theoretical results, which facilitates the task of the reader.

 * One minor weakness is that the paper does not allocate much effort to clarifying the practical implications of the proposed solution. It sounds intuitively nice to have training-free risk certificates for a model but the text will be improved if the paper gives one or two down-to-earth examples about what one can do with them in real life.

---

> ### Author Response · Authors · 2023-04-02
> **Rebuttal by Authors**
>
> Thank you for your time and your positive feedback! Please see our detailed responses to your comments and suggestions below.
>
> **Q1** There may be a typo on the r.h.s. of Eq (4). Should there really be a multiplicative empirical risk term there.
>
> We assert that there is no typographical error on the right-hand side (r.h.s.) of Eq (4). Here is a detailed explanation of Eq (4), derived from Eq (3), which is an inequality:
>
> ${\rm kl}\big( {R}_{\mathcal{S}}(Q) | {R}_\mathcal{D} (Q) \big) \leq \frac{ \mathrm{KL} (Q | Q(0)) + \log \frac{2\sqrt{n}}{\delta}} {n}$
>
> Applying Pinsker's inequality for binary KL divergence, ${\rm kl}(\hat p | p) \ge (p-\hat p)^2/(2p)$, where $\hat p < p$, we can further expand Eq (3) into Eq(4):
>
> $R_\mathcal{D}(Q) - {R}_{\mathcal{S}}(Q) \leq \sqrt{ 2R_\mathcal{D}(Q) \frac{ \mathrm{KL} (Q | Q(0)) + \log \frac{2\sqrt{n}}{\delta}} {n}}.$
>
> where we take $p = R_\mathcal{D}(Q)$ and $\hat{p}= R_\mathcal{S}(Q)$.
>
> **Q2** While the PAC Bayes aspect is a clear novelty, generalization behavior of NTKs has been previously analyzed. The paper will be more complete if some of this body of work is cited and the advantages of the PAC-Bayesian framework on the state of the art there are clarified. For instance:
>
> M. Nonnenmacher et al., Which Minimizer Does My Neural Network Converge To?, ECML-PKDD, 2021
>
> Thank you for introducing this relevant work. We have now cited it, along with other studies that use kernel regression to characterize the generalization behavior of standard neural networks. Please refer to our updated manuscript, where the changes have been marked in blue for your convenience.

---

### Review · Reviewer_Q6fj · 2023-03-25

**Summary Of Contributions:**

The authors study training and generalization of infinite width deep neural networks "trained by" gradient descent, by minimizing an objective that is based on a PAC-Bayes bound with Gaussian priors/posteriors. They extend existing theoretical results for infinite width neural networks to probabilistic networks, where the weights are being sampled from a Gaussian, and the mean of the Gaussian is being trained. They then characterize convergence of the appropriate Gram matrix, as well as generalization of the average output of this randomized predictor. Some empirical results and evaluation of the bound is presented. The authors also propose a way to select hyperparameters (effectively adjusting the amount of regularization coming from the PAC-Bayes bound).

**Audience:**

Yes

**Broader Impact Concerns:**

No concerns.

**Claims And Evidence:**

No

**Requested Changes:**

1. I would appreciate the authors clearing up the confusion I see between Gibbs risk and risk of average predictor at equation (8). Please, edit the text to carefully deal with these quantities, highlighting the differences and using different notation to capture these differences ( see my comment under weaknesses). The loss being considered should be specified before any derivation based on it. If (8) requires derivation, it should be given. Known bounds relate average predictor risk to Gibbs risk (a factor of 2 suffices; see work on PAC-Bayes "C-bounds" for more sophisticated approaches).

2. The results and proofs appear to be very connected to those existing in the literature for deterministic networks (see my comment above under weaknesses), and thus including a discussion around this seems prudent.


3. The second bullet point in the contributions seems to suggest that the bound is independent of the posterior. While explicitly it is true, isn’t the bound valid under a Gaussian posterior, where only the mean parameter is being trained? As written, the contribution statement seems to be at the very least confusing, but potentially either inaccurate or misleading.

4. Based on the paragraph above 3.2, it looks like the covariance matrix is restricted to be diagonal, or even isotropic? It might be worth adding a comment here for the readers. In particular, late in the experiment section, the authors note that they only train the mean and not the variance, and show some empirical results to justify (Section C.2). However, such results may be highly affected by the fact that the authors do not model the correlations between the weights. Also, just because the gradients of the variance parameters are much smaller compared to mean parameter updates, it does not mean that the changes in the variance are insignificant. Perhaps adjusting the learning rate for variance parameters or applying other learning tricks could allow the variance to be learnt more effectively. Could the authors comment more on this observation and provide further justification for their choice to only train the mean parameters?

5. In Equation 9: it seems like the learning rate is missing. In particular, I believe the authors use gradient flow dynamics to replace \dee theta / \dee t with a gradient update, but this part requires the learning rate to appear. I suppose it does not really matter since no discretization of the dynamics is going on, but it would be good to clarify.

6. In Section 5.1, the authors mention that they focus on the kernel associated to the final layer, and say “at the cost of a minor degradation in convergence rates”. I did not get this comment and do not obviously see how the convergence rate is affected by dropping the rest of the terms, could the authors expand?

Minor comments and citation edits:
 - In the introduction, 1st paragraph, seems like it would be natural to cite Sharpness Aware Minimization (SAM) work;
 - Top of page 2 “...making PAC-Bayes learning an essential tool..” seems like an overstatement. In particular, I do not think that PAC-Bayes is standard practice in training best-performing models.
 - Dziugaite et al. 2020 work focuses on showing that distribution and data-dependent priors is in some cases are necessary to get tight generalization guarantees and perform better than just distribution-dependent priors. This work seems to be mis-cited (top of page 1, and missing in Section 6.1 where data-dependent prior is being justified).
 - Related work first paragraph, along the Haddouche et al 2021 citation, i believe one should cite Germain et al. 2016 paper (PAC-Bayesian theory meets Bayesian inference) where they provide bounds for sub-gamma loss functions.
 - Page 4, just above equation two, the authors say “To ensure that the weights follow a Gaussian distribution during gradient descent training, we use the re-parameterization trick”. While I understand what the authors meant, this initially confused me. I think it is worth highlighting instead that _based on_ the re-parameterization, the weights are being modeled as Gaussian, with tranable mean and variance parameters.
 - While the meaning of X in Eq 8 is obvious, I believe it may be not defined.
 - Typo: first paragraph in Sec 6 (a empirical)
 - 6.2 first paragraph on grid search, might be worth noting that there is a “penalty” associated with hyperparameter search, though relatively small.
 - Typo: 5.3 first paragraph, “consists two terms”


**Strengths And Weaknesses:**

I have some concerns regarding the definitions made in the preliminaries (see below), which for now determined my negative response to the *Claims And Evidence* question below.

*Randomized predictor vs average predictor*

My main concern / confusion is around the (empirical) risk of a randomized predictor, versus the risk of the average predictor: both seem to be used, and potentially a bound on one quantity is incorrectly applied to a PAC-Bayes bound, which uses another quantity (at least without an explanation).

In 3.2, the authors denote the "expected population risk" (what is often referred to as Gibbs risk) by R_D(Q), defined as the expectation of risk_D(h) when h~Q. Similarly, they denote the expected empirical risk by R_S(Q). The main theoretical results is stated in terms of the Gibbs Risk R_D(Q), but in equation (8), they state that R_S(Q) is equal to the risk of the average predictor , risk (E(h)). By Jensen's, I would expect equation (8) to be expressing a lower bound on the Gibbs empirical risk, not an equality.

In fact, risk are defined in terms of expected losses, but we arrive at (8) without any mention of the underlying loss, which only suppose is 1/2 squared error. It would be nice for readers to not have to intuit the origins of the equality given in (8). One concern is that a lower bound is being used and combined with an upper bound, which in general yields no bound at all. I'm assuming there's a misunderstanding, but it is critical to address this.

*Proof techniques and results relationship to previous work*

Similar results for deterministic networks exist in the literature. Intuitively, it seems like those results would straightforwardly extend to probabilistic networks when the average output is being studied (instead of a randomized predictor, see my comment above).

How do the proofs presented in this paper (e.g., for Thm 4.2) relate to those for NTK in deterministic networks? If they are basically the same, or could be made the same, then the authors should cite appropriate work that their proofs are following. If not, then I think it would be worth highlighting what the obstacles are and why studying the average output of a randomized network is more challenging or different.

In particular, the results in Theorem 4.3 (since only the mean parameter is being updated, as in Eq 14) seem to be a fairly straightforward consequence of standard NTK results (and the existing relationship between PAC-Bayes with Gaussian priors/posteriors and ridge regression). If not, what are the key differences?

---

> ### Author Response · Authors · 2023-04-02
> **Rebuttal by Authors Part I**
>
> Thanks for your time and your detailed comments that help to improve our work. Please see our detailed responses to your comments and suggestions below, where we reorganize the weakness and questions a bit to better clarify your concerns. The response will be a bit long and we sincerely appreciate your time and patience. Please refer to our updated manuscript, where the changes have been marked in blue for your convenience.
>
> **W1** Randomized predictor vs average predictor. My main concern / confusion is around the (empirical) risk of a randomized predictor, versus the risk of the average predictor: both seem to be used, and potentially a bound on one quantity is incorrectly applied to a PAC-Bayes bound, which uses another quantity (at least without an explanation).
>
> We appreciate your concern regarding the distinction between the risk of a randomized predictor and the risk of the average predictor. We have defined the empirical risk in Equation (8) as follows:
>
> $ \underline{R}_{\mathcal{S}} (Q;t) = \frac{1}{2n} \left || \mathbf{y} - \mathbb{E}\_{f \sim Q} f(\mathbf{X};t) \right ||^2_2 = \frac{1}{2n}  || \mathbf{y} - \hat{f}(\mathbf{X}; t) ||^2_2,$
>
> Our choice of empirical risk is inspired by the PAC-Bayes $\mathcal{C}$-bound, which serves as an upper bound for the risk of Bayes classifiers. In this work, we consider an empirical loss that represents a mean squared error (MSE) version of the Bayes classifiers."
> This choice has the advantage of providing an explicit solution for the expected output function when considering the infinite-width limit. For additional details, kindly refer to our response to C1.
>
> **W2&C2** Proof techniques and results relationship to previous work
> Similar results for deterministic networks exist in the literature. Intuitively, it seems like those results would straightforwardly extend to probabilistic networks when the average output is being studied (instead of a randomized predictor, see my comment above).
>
> Our proof framework is inspired by and follows the approach presented in [1], particularly sharing the same three core steps:
>
> - Show at initialization $\lambda_{0} \left(\boldsymbol{\Theta}^{(L)}(0) \right) \ge \frac{\lambda_0 \left(\mathbf{K}_\infty^{(L)} \right)}{2} $ and the required condition on $m$.
>
> - Show during training $\lambda_{0} \left(\boldsymbol{\Theta}^{(L)}(t) \right) \ge \frac{\lambda_0 \left(\mathbf{K}_\infty^{(L)} \right)}{2} $ and the required condition on $m$.
>
> - Show during training the expected empirical loss $R_\mathcal{S}(Q)$ has a linear convergence rate.
> \end{itemize}
>
> However, our network architecture is considerably more complex. The probabilistic network contains two sets of parameters, resulting in two NTKs. Consequently, our proof demands more elaborate bounding of numerous terms. For instance, in Lemma A.3 and Lemma A.5, we need to bound the PNTK associated with the variance.
>
> We have incorporated the aforementioned clarification in our revised manuscript.
>
> **C1** I would appreciate the authors clearing up the confusion I see between Gibbs risk and risk of average predictor at equation (8). Please, edit the text to carefully deal with these quantities, highlighting the differences and using different notation to capture these differences ( see my comment under weaknesses). The loss being considered should be specified before any derivation based on it. If (8) requires derivation, it should be given. Known bounds relate average predictor risk to Gibbs risk (a factor of 2 suffices; see work on PAC-Bayes "C-bounds" for more sophisticated approaches).
>
> Thank you for seeking clarification on Equation (8) and the suggestion of the PAC-Bayes C-bound. We have made the following changes as requested:
>
> - We have provided an introduction to the PAC-Bayes C-bound in Section 3.2, including relevant definitions and generalization bounds based on references [2,3,4].
>
> -  In Section 3.3, we have explicitly demonstrated the training objective function:
>
> $L(Q)  = \underline{R}_\mathcal{S}(Q) + \lambda \frac{\mathrm{KL}( \boldsymbol{\theta}(t) \| \boldsymbol{\theta}(0)) }{n} =  \frac{1}{2n} \left || \mathbf{y} - \mathbb{E}\_{f \sim Q} f(\mathbf{X};t) \right ||^2_2 + \lambda \frac{\mathrm{KL}( \boldsymbol{\theta}(t) \| \boldsymbol{\theta}(0)) }{n} $
>
> In this work, we employ the empirical loss of the expected output function and set ℓ to be the squared loss in the training objective function. This approach offers the advantage of yielding an explicit solution for the expected output function in the infinite-width limit.
>
> -  We have modified Equation (8) (now Equation 9 in the revised manuscript) to explicitly show the empirical loss.
> - We have updated Theorem 4.4 accordingly.

---

> ### Author Response · Authors · 2023-04-02
> **Rebuttal by Authors Part II**
>
> **C3**  The second bullet point in the contributions seems to suggest that the bound is independent of the posterior. While explicitly it is true, isn’t the bound valid under a Gaussian posterior, where only the mean parameter is being trained? As written, the contribution statement seems to be at the very least confusing, but potentially either inaccurate or misleading.
>
> Thank you for pointing that out. We have updated the statement to clarify the relationship between the bound and the posterior. Here is our revised statement:
>
> Building upon the optimization solution, we derive an analytical and guaranteed PAC-Bayesian bound for deep networks after training for the first time. In contrast to other PAC-Bayesian bounds that require the distribution of the posterior, our bound, based on the $\mathcal{C}$-bound under a Gaussian posterior, where only the mean parameter is being trained, is entirely independent of computing the distribution of the posterior.
>
> **C4** Based on the paragraph above 3.2, it looks like the covariance matrix is restricted to be diagonal, or even isotropic? It might be worth adding a comment here for the readers. In particular, late in the experiment section, the authors note that they only train the mean and not the variance, and show some empirical results to justify (Section C.2). However, such results may be highly affected by the fact that the authors do not model the correlations between the weights. Also, just because the gradients of the variance parameters are much smaller compared to mean parameter updates, it does not mean that the changes in the variance are insignificant. Perhaps adjusting the learning rate for variance parameters or applying other learning tricks could allow the variance to be learnt more effectively. Could the authors comment more on this observation and provide further justification for their choice to only train the mean parameters?
>
> We would like to clarify that in our experiments, we train both the mean and variance parameters. The assumption of fixed variance is used only in our theoretical analysis.
>
> We have added a comment for readers at the end of Section 3.1:
> Our assumption of an isotropic initialization for the covariance matrix is grounded in the empirical setting presented in [5].
>
> **C5**  In Equation 9: it seems like the learning rate is missing. In particular, I believe the authors use gradient flow dynamics to replace \dee theta / \dee t with a gradient update, but this part requires the learning rate to appear. I suppose it does not really matter since no discretization of the dynamics is going on, but it would be good to clarify.
>
> Thank you for pointing it out. We have included an explanation in the updated manuscript:
>
> Without loss of generality, we set the learning rate $\eta = 1$.
>
> **C6**  In Section 5.1, the authors mention that they focus on the kernel associated to the final layer, and say “at the cost of a minor degradation in convergence rates”. I did not get this comment and do not obviously see how the convergence rate is affected by dropping the rest of the terms, could the authors expand?
>
> The definition of NTK is given by Equation 11:
>
> $\boldsymbol{\Theta}\_{\mu}({\bf X},{\bf X};t) = \frac{\partial \hat{f}(\mathbf{X};t)}{\partial \boldsymbol{\theta}\_\mu} \left(\frac{\partial \hat{f}(\mathbf{X};t)}{\partial \boldsymbol{\theta}\mu} \right)^\top = \sum_{l=1}^L \nabla_{ \mathbf{W}\mu^{(l)} } \hat{f}({\bf X};t) \nabla_{ \mathbf{W}_\mu^{(l)} } \hat{f}({\bf X};t)^{\top}$
>
> This is a summation of the NTK at each layer. Using the inequality $\lambda_0 (\boldsymbol{\Theta}\_{\mu}^{(L)}) < \lambda_0(\boldsymbol{\Theta}\_{\mu})$, we obtain the convergence result, albeit with a minor degradation in convergence rates.
>
> **M1** In the introduction, 1st paragraph, seems like it would be natural to cite Sharpness Aware Minimization (SAM) work;
>
> Thank you for your suggestion. We have cited the SAM work [6] in the first paragraph of our updated manuscript.
>
> **M2** Top of page 2 “...making PAC-Bayes learning an essential tool..” seems like an overstatement. In particular, I do not think that PAC-Bayes is standard practice in training best-performing models.
>
> Thank you for your valuable feedback. While it is true that PAC-Bayesian learning might not currently be considered "standard practice" in training best-performing models, it is important to recognize its potential contributions to the field of deep learning. We have changed "essential" to "valuable" to convey the importance of PAC-Bayesian learning without implying that it is absolutely necessary.

---

> ### Author Response · Authors · 2023-04-02
> **Rebuttal by Authors Part III**
>
> **M3** Dziugaite et al. 2020 work focuses on showing that distribution and data-dependent priors is in some cases are necessary to get tight generalization guarantees and perform better than just distribution-dependent priors. This work seems to be mis-cited (top of page 1, and missing in Section 6.1 where data-dependent prior is being justified).
>
> Thanks for pointing it out. We have moved the citation to the top of page 2 and added a citation of Dziugaite et al. 2020 [7] in Section 6.1.
>
> **M4** Related work first paragraph, along the Haddouche et al 2021 citation, i believe one should cite Germain et al. 2016 paper (PAC-Bayesian theory meets Bayesian inference) where they provide bounds for sub-gamma loss functions.
>
> Thanks for your suggestion, we have cited Germain et al. 2016 [8].
>
> **M5** Page 4, just above equation two, the authors say “To ensure that the weights follow a Gaussian distribution during gradient descent training, we use the re-parameterization trick”. While I understand what the authors meant, this initially confused me. I think it is worth highlighting instead that based on the re-parameterization, the weights are being modeled as Gaussian, with tranable mean and variance parameters.
>
> Thanks for your suggestion, we have modified the statement as follows:
>
> We employ the reparameterization trick to model the weights as Gaussian during gradient descent training, with trainable mean and variance parameters.
>
> **M6**  While the meaning of X in Eq 8 is obvious, I believe it may be not defined.
>
> Thanks for pointing it out, we added a definition of X.
>
> **M7** Typo: first paragraph in Sec 6 (a empirical)
>
> Thanks for pointing it out, we have revised.
>
> **M8** 6.2 first paragraph on grid search, might be worth noting that there is a “penalty” associated with hyperparameter search, though relatively small.
>
> We incorporated a sentence that acknowledges the "penalty" associated with hyperparameter search while emphasizing its relatively small impact:
>
> While it is worth noting that there is a ``penalty'' associated with $\delta$ (see Equation 6), this penalty is typically relatively small compared to the potential performance improvements that can be gained through optimal hyperparameter selection.
>
> **M9** Typo: 5.3 first paragraph, “consists two terms”
>
> Thanks for pointing it out, we have revised.
>
>
> **References**
>
> [1] Du, Simon, et al. "Gradient descent finds global minima of deep neural networks." International conference on machine learning. PMLR, 2019.
>
> [2] Laviolette, François, and Mario Marchand. "PAC-Bayes Risk Bounds for Stochastic Averages and Majority Votes of Sample-Compressed Classifiers." Journal of Machine Learning Research 8.7 (2007).
>
> [3] Laviolette, François, Mario Marchand, and Jean-Francis Roy. "From PAC-Bayes bounds to quadratic programs for majority votes." Proceedings of International Conference on Machine Learning. 2011.
>
> [4] Germain, Pascal, et al. "Risk bounds for the majority vote: From a PAC-Bayesian analysis to a learning algorithm." arXiv preprint arXiv:1503.08329 (2015).
>
> [5] Rivasplata, Omar, Vikram M. Tankasali, and Csaba Szepesvári. "PAC-Bayes with backprop." arXiv preprint arXiv:1908.07380 (2019).
>
> [6] Foret, Pierre, et al. "Sharpness-aware minimization for efficiently improving generalization." arXiv preprint arXiv:2010.01412 (2020).
>
> [7] Dziugaite, Gintare Karolina, et al. "On the role of data in PAC-Bayes bounds." International Conference on Artificial Intelligence and Statistics. PMLR, 2021.
>
> [8] Germain, Pascal, et al. "PAC-Bayesian theory meets Bayesian inference." Advances in Neural Information Processing Systems 29 (2016)

---

> > ### Comment · Reviewer_Q6fj · 2023-04-23
> > **A couple more points**
> >
> > Thanks for answering my questions and making edits to the paper. A couple more things:
> > 1) I looked through the edits. Now the C-bound is mentioned many times early on, but it's used like undefined jargon. I would also have an explicit bound with a name C-bound so that the readers could actually find it. I would also add an explanation in the intro, not just later in text.
> > 2) I agree with another reviewer that Dziugaite and Roy '17 should be more prominently cited and compared to due to the similarity of the objectives.

---

> > > ### Author Response · Authors · 2023-04-25
> > > **Author Responses to Further Questions**
> > >
> > > We appreciate your additional comments and suggestions. Our responses are provided below:
> > >
> > > **Q1** I looked through the edits. Now the C-bound is mentioned many times early on, but it's used like undefined jargon. I would also have an explicit bound with a name C-bound so that the readers could actually find it. I would also add an explanation in the intro, not just later in text.
> > >
> > > **A1**  We have now incorporated the explicit bound into Theorem 3.3 and Equation (6) in the revised manuscript:
> > >
> > > $ R\_\mathcal{D} (B\_Q)   \le  1 - \frac{ (\mathcal{M}^{\mathcal{D}}\_Q)^2 }{ \mathcal{M}^{\mathcal{D}}\_{Q^2}}  \leq 1 - \frac{(\mathcal{M}^\mathcal{S}\_Q - 2B \frac{\sqrt{\ln(\frac{2n}{\delta})}}{\sqrt{2n}} )^2}{\mathcal{M}^\mathcal{S}\_{Q^2} + 2B^2 \frac{\sqrt{\ln(\frac{2n}{\delta})}}{\sqrt{2n}} }$.
> > >
> > > Besides, we have included an explanation in the Introduction section:
> > >
> > > The PAC-Bayes bounds employed in Deep PAC-Bayesian learning typically stem from the PAC-Bayes-kl theorem, as indicated by [1,2]. These bounds study the expected risk, often referred to as Gibbs risk. Conversely, an alternative line of PAC-Bayes bounds, known as the $\mathcal{C}$-bound [3], presents an upper bound for the risk associated with the Majority Vote classifier, the risk of expected output. Examining the risk of expected output within the PAC-Bayes bound presents the benefit of providing an explicit solution for optimization analysis in the infinite-width limit. Moreover, we formulate the generalization bound of PAC-Bayesian learning upon convergence for the first time.
> > >
> > > [1] Langford, John, and Matthias Seeger. Bounds for averaging classifiers. School of Computer Science, Carnegie Mellon University, 2001.
> > >
> > > [2] Seeger, Matthias. "PAC-Bayesian generalisation error bounds for Gaussian process classification." Journal of machine learning research 3.Oct (2002): 233-269.
> > >
> > > [3] Lacasse, Alexandre, et al. "PAC-Bayes bounds for the risk of the majority vote and the variance of the Gibbs classifier." Advances in Neural information processing systems 19 (2006).
> > >
> > > **Q2** I agree with another reviewer that Dziugaite and Roy '17 should be more prominently cited and compared to due to the similarity of the objectives
> > >
> > > **A2** Thank you for your recommendation. We have incorporated the discussion related to Dziugaite and Roy '17 in the revised manuscript:
> > >
> > > Contrasting the generalization bound presented by [Dziugaite and Roy '17], which computes the average over the training set using random samples, our approach utilizes the empirical loss of the expected output function inspired by the PAC-Bayes $\mathcal{C}$-bound. In particular, the empirical loss $ \underline{R}\_\mathcal{S}(Q) \triangleq \frac{1}{n} \sum_{i=1}^{n} \ell \left( y_{i}, \hat{f} \left( {\bf x}{i} \right) \right) $ used in Equation (7) is a lower bound of the expected empirical loss (also known as Gibbs empirical risk) $R_\mathcal{S}(Q)$, as per Jensen's inequality. This approach offers the advantage of yielding an explicit solution for the expected output function in the infinite-width limit.
> > >
> > > **Note that all updates have been highlighted in purple.**

---

### Review · Reviewer_c5C9 · 2023-04-05

**Summary Of Contributions:**

This paper studies the gradient descent dynamics of PAC-Bayesian learning in wide neural networks using the NTK framework. The authors identify a modified NTK, called Probabilistic NTK (PNTK), which accounts for the probabilistic nature of NNs in the PAC-Bayesian framework, as well as the fact that such settings have two sets of parameters: one for parameter mean and one for variances. Using this, the authors derive theoretical contributions including convergence analysis (which is shown to be kernel ridge regression using PNTK *if* one assumes that variance parameters are fixed), as well as generalisation analysis for probabilistic NNs. Finally, the authors use their bounds to suggest a method for automatic hyperparameter selection without needing to train, and demonstrate empirically the efficacy of their method compared to some standard baselines for hyperparameter selection.

**Audience:**

Yes

**Claims And Evidence:**

Yes

**Requested Changes:**

In addition to addressing/commenting on my above weaknesses, I'd appreciate it if the authors can clarify the following clarity concerns I have:

1. What is a $\textit{C}$-bound? This is introduced without explanation.
2. I don't think the name "Pac-Bayes bound score", for the hparam selection method, is introduced until the conclusion. I would either not include it or introduce earlier.
3. I would put section 2 somewhere after section 3, because it introduces things like prior and posterior in PAC-Bayesian setting which someone without previous knowledge would have access to.
4. In section 3.2, does the loss function need to be bounded in order for Theorem 3.1 to apply? As currently written, this isn't clear (though the binary KL divergence doesn't make sense if loss isn't in [0,1]).

**Strengths And Weaknesses:**

*Strengths*

1. The paper tackles an interesting problem of analysing the behaviour of PAC-Bayesian trained NNs in the wide NN setting through the NTK.
2. The authors use their theory to derive a practical way to automate the hyperparameter selection of NNs.
3. The paper is reasonably well written and clear.

*Weaknesses*

1. It's not clear to me that the authors have shown that the NTK kernel and training dynamics limit *exists* for the PAC-Bayesian setting when the variance parameters exist in addition to the mean parameters. I think this is potentially an interesting area, but this seems different from existing NTK works, and would needs more of a rigorous treatment to be convincing to me. For example, the authors should write out explicitly the analytic forms of $\Theta_{\mu}$ and $\Theta_{\sigma}$ over different layers in an MLP, especially $\Theta_{\sigma}$ which is new as far as I can tell.
2. I have some confusions regarding the theoretical contributions, which seem to be somewhat simplified, to the extent that it's not clear the extent to which the contributions are providing new interesting insights into the NTK regime behaviour of PAC-Bayesian NNs. For example:
a. it is not clear to me why theorem 4.2 is stated in terms of the NNGP not the NTK; the NNGP is a kernel that is obtained via training the last layer *only*, and treats the hidden layers solely as data-independent feature extractors.
b. the variance parameters are treated as fixed in theorem 4.3, which I assume is for simplicity, but also reduces the analysis down to standard Kernel RR with mean parameter NTK (which is very similar to the non-probabilistic case).
c. On this note, the function mean at initialisation is assumed to be zero in theorem 4.3 (which is important to obtain kernel RR, rather than a random GP as in Lee et al 2019, or He et al 2020 [https://arxiv.org/abs/2007.05864](https://arxiv.org/abs/2007.05864)).
d. It would also be good to have some explanation why in Appendix A the authors consider the expected NN averaged over the PAC-Bayes randomness (e.g. in equation 19). Presumably this isn't what is being used in practice?

2. In the experiments, it would be good to have the actual test accuracies and cross entropies in Table 1, in order to determine the tightness of the proposed bounds.
3. The Scability of the proposed "PAC-Bayesian bound score" for hyperparameter selection seems poor to me, due to the vector-Matrix inverse-vector products in equation 18. Can the authors comment on how best to compute this efficiently, and/or if this should be a concern? For example, can the authors provide experiments that use different amounts of the training data to compute the bounds for hparam selection, and assess if one can get away with using a small fraction of the training data to do so?

---

> ### Author Response · Authors · 2023-04-18
> **Rebuttal by Authors Part I**
>
> Thank you for your time and your positive feedback! Please see our detailed responses to your comments and suggestions below.
>
> **W1** It's not clear to me that the authors have shown that the NTK kernel and training dynamics limit exists for the PAC-Bayesian setting when the variance parameters exist in addition to the mean parameters. I think this is potentially an interesting area, but this seems different from existing NTK works, and would needs more of a rigorous treatment to be convincing to me. For example, the authors should write out explicitly the analytic forms of $\Theta_\mu$ and $\Theta_\sigma$ over different layers in an MLP, especially $\Theta_\sigma$, which is new as far as I can tell.
>
> **A1** Thank you for your suggestion. We have now explicitly provided the analytic expressions for both $\Theta_\mu$ and $\Theta_\sigma$ as follows:
>
> $ \boldsymbol{\Theta}\_{\mu,ij}^{(l)}(0)  =  \sum\_{r=1}^m \mathbb{E}\_{\boldsymbol{\xi}} \left[ \frac{1}{\sqrt{m}} \frac{\partial \hat{f}(\mathbf{x}\_i) }{\partial {x}^{(l)}\_{i,r} }  \sigma'(( {\bf w}\_{r}^{(l)})^\top {\bf x}\_i^{(l-1)} ) \right] \mathbb{E}\_{\boldsymbol{\xi}} \left[ \frac{1}{\sqrt{m}} \frac{\partial \hat{f}(\mathbf{x}\_j) }{\partial {x}^{(l)}\_{j,r} }  \sigma'(( {\bf w}\_{r}^{(l)})^\top  {\bf x}\_j^{(l-1)} ) \right]$
>
> $\boldsymbol{\Theta}\_{\sigma,ij}^{(l)}(0) =  \sum\_{r=1}^m \mathbb{E}\_{\boldsymbol{\xi}} \left[ \frac{1}{\sqrt{m}} \frac{\partial \hat{f}(\mathbf{x}\_i) }{\partial {x}^{(l)}\_{i,r} }  \sigma'( ({\bf w}\_{r}^{(l)})^\top \hat{\bf x}\_i^{(l-1)}) \odot \boldsymbol{\xi}^{(l)}\_r \right] \mathbb{E}\_{\boldsymbol{\xi}} \left[ \frac{1}{\sqrt{m}} \frac{\partial \hat{f}(\mathbf{x}\_i) }{\partial {x}^{(l)}\_{j,r} }  \sigma'( ({\bf w}\_{r}^{(l)})^\top {\bf x}\_j^{(l-1)} ) \odot \boldsymbol{\xi}^{(l)}\_r \right].$
>
> We have also revised the manuscript to include the updated explicit expressions for the Probabilistic Neural Tangent Kernel (PNTK). Please refer to the changes highlighted in red for your convenience.
>
> Building upon the explicit expressions, we demonstrate that both PNKTs remain near their initial values throughout the training process, thereby establishing a convergence theory for PNNs with large widths. Our proof is constructed using three key steps, as outlined below:
>
>   - Show at initialization $\lambda_{0} \left(\boldsymbol{\Theta}^{(L)}(0) \right) \ge \frac{\lambda_0 \left(\mathbf{K}_\infty^{(L)} \right)}{2} $ and the required condition on $m$.
>
>  - Show during training $\lambda_{0} \left(\boldsymbol{\Theta}^{(L)}(t) \right) \ge \frac{\lambda_0 \left(\mathbf{K}_\infty^{(L)} \right)}{2} $ and the required condition on $m$.
>
>   - Show during training the expected empirical loss $R_\mathcal{S}(Q)$ has a linear convergence rate.
>
> **W2** I have some confusions regarding the theoretical contributions, which seem to be somewhat simplified, to the extent that it's not clear the extent to which the contributions are providing new interesting insights into the NTK regime behaviour of PAC-Bayesian NNs. For example: a. it is not clear to me why theorem 4.2 is stated in terms of the NNGP not the NTK; the NNGP is a kernel that is obtained via training the last layer only, and treats the hidden layers solely as data-independent feature extractors. b. the variance parameters are treated as fixed in theorem 4.3, which I assume is for simplicity, but also reduces the analysis down to standard Kernel RR with mean parameter NTK (which is very similar to the non-probabilistic case). c. On this note, the function mean at initialisation is assumed to be zero in theorem 4.3 (which is important to obtain kernel RR, rather than a random GP as in Lee et al 2019, or He et al 2020 https://arxiv.org/abs/2007.05864). d. It would also be good to have some explanation why in Appendix A the authors consider the expected NN averaged over the PAC-Bayes randomness (e.g. in equation 19). Presumably this isn't what is being used in practice?
>
> **A2**  a) In this paper, we primarily focus on the PNTK at the last layer, represented by $\boldsymbol{\Theta}^{(L)} = \boldsymbol{\Theta}^{(L)}\_\mu + \boldsymbol{\Theta}^{(L)}\_\sigma$. This approach simplifies the Gram matrix induced by the weights from the $L$-th layer, albeit at the expense of a minor degradation in convergence rate. Specifically, as $\Theta = \sum^L_{l=1} \Theta^{(l)}$, it follows that $\lambda_\min(\Theta^L) < \lambda_\min(\sum^L_{l=1} \Theta^{(l)}) = \lambda_\min(\Theta)$.
>
> Moreover, based on the relationship between NNPG ($\mathbf{K}$) and PNTK at the last layer, we can establish an upper bound for the PNTK at the last layer:
>
> $\left| \boldsymbol{\Theta}^{(L)}\_\mu - \mathbf{K}^{(l)}\_\infty \right|_2 \le \sum\_{ij} \left | \boldsymbol{\Theta}^{(L)}\_{\mu,ij} - \mathbf{K}^{(l)}\_{ij,\infty} \right| \le C^{L} n \sqrt{ \frac{\log (Ln /\delta)}{m}}$.

---

> ### Author Response · Authors · 2023-04-18
> **Rebuttal by Authors Part II**
>
> **A2** b) We consider the variance parameters to be fixed for two main reasons:
>
> During training, we empirically observe that the norm of $\frac{\partial f}{\partial \theta_\sigma}$ is significantly smaller than $\frac{\partial f}{\partial \theta_\mu}$. This suggests that the variance weights can be regarded as approximately fixed during the training process. We present this empirical evidence in Appendix C.3 (Figure 4).
>
> We also take into account the analytic theoretical result. When the variance weights are fixed, the objective function with PNTK leads to kernel ridge regression. This analytic outcome has the benefit of providing an explicit solution for the PNN, which can be further leveraged to characterize the generalization bound."
>
> c) Thank you for highlighting this point. We concur that we employ the same strategy, assuming the function mean at initialization to be zero.
>
> d) We compute the derivative of the expected output function with respect to the parameters based on the definition of the objective function. It is important to note that in practice, during (stochastic) gradient descent training, we do not take the expected NN average over the PAC-Bayes randomness. This distinction arises from the difference between the definition of empirical loss used in the PAC-Bayes bound and the actual training process.
>
> **W3** In the experiments, it would be good to have the actual test accuracies and cross entropies in Table 1, in order to determine the tightness of the proposed bounds.
>
> **A3** Thank you for your suggestion. We have now included test accuracies (acc.) and test cross-entropy errors (x-e) in Table 1:
>
> |             Data            |         Method          | Network | $\ell^{x-e}$ | $\ell^{01}$ | x-e | acc. |  Single  |   Total   |
> |:---------------------------:|:-----------------------:|:-------:|:------------:|:-----------:|:--------------------:|:--------------------:|:--------:|:---------:|
> |          **MNIST**          |   Exhaustive Search     |   FCN   |    .0010     |    .0212    |         .0001        |         .0189        |   .50    |   324.00  |
> |                             |                         |   CNN   | **.0006**    | **.0110**   |         .0004        |         .0093        |  16.92   | 10964.16  |
> |                             |     Bayesian Search     |   FCN   |    .0010     |    .0212    |         .0001        |         .0189        |   .50    |  **18.00**|
> |                             |                         |   CNN   | **.0006**    | **.0110**   |         .0004        |         .0093        |  16.92   |  609.12   |
> |                             |        $\mathcal{PA}$   |   FCN   |    .0011     |    .0264    |         .0002        |         .0208        |   .03    |  19.44    |
> |                             |                         |   CNN   |  **.0009**   |  **.0160**  |         .0008        |         .0108        |   .03    |  19.44    |
> |         **CIFAR10**         |   Exhaustive Search     |   FCN   |    .1740     |    .5377    |         .0051        |         .4866        |   1.09   |  706.32   |
> |                             |                         |   CNN   | **.0142**    | **.1969**   |         .0023        |         .1510        |  45.00   | 29160.00  |
> |                             |     Bayesian Search     |   FCN   |    .1740     |    .5377    |         .0051        |         .4866        |   1.09   |  39.24    |
> |                             |                         |   CNN   | **.0142**    | **.1969**   |         .0023        |         .1510        |  45.00   | 1620.00   |
> |                             |        $\mathcal{PA}$   |   FCN   |    .1780     |    .5490    |         .0048        |         .4920        |   .03    |  **19.44**|
> |                             |                         |   CNN   | **.0142**    |  **.1970**  |         .0024        |         .1511        |   .03    |  **19.44**|
>
> The results clearly show that the proposed bound is effectively tight.
>
> **W4**  The Scability of the proposed "PAC-Bayesian bound score" for hyperparameter selection seems poor to me, due to the vector-Matrix inverse-vector products in equation 18. Can the authors comment on how best to compute this efficiently, and/or if this should be a concern? For example, can the authors provide experiments that use different amounts of the training data to compute the bounds for hparam selection, and assess if one can get away with using a small fraction of the training data to do so?
>
> **A4** We select a subset of data to strike a balance between computational efficiency and accuracy. Our experimental results indicate that when the size of the data subset exceeds a certain threshold (125 samples per class), the performance of PA with varying hyperparameters stabilizes. Please refer to our updated simulation results in Appendix C.6.

---

> ### Author Response · Authors · 2023-04-18
> **Rebuttal by Authors Part III**
>
> **C1.** What is a C-bound? This is introduced without explanation.
>
> **R1** The C-bound serves as an upper bound for the risk of the Majority Vote classifier, relying on the first two moments of the margin of the Q-convex combination realized on the data. On the other hand, the PAC-Bayes bound (Theorems 3.1, 3.2) focuses on classifiers that utilize a fixed subset of training examples. In contrast, the C-bound applies to a stochastic average (and a majority vote) of classifiers using varying subsets (with different sizes) of training examples. Specifically, the C-bound is designed for the limited case where the voters are all classifiers, aligning with the kernel ridge regression derived in this work. We have updated our manuscript to reflect this information.
>
> **C2** I don't think the name "Pac-Bayes bound score", for the hparam selection method, is introduced until the conclusion. I would either not include it or introduce earlier.
>
> **R2** Thank you for highlighting this issue. We have now removed the "PAC-Bayes bound score" from the text.
>
> **C3**  I would put section 2 somewhere after section 3, because it introduces things like prior and posterior in PAC-Bayesian setting which someone without previous knowledge would have access to.
>
> **R3** Thank you for your suggestion. We would like to clarify that the current arrangement of placing Section 3.2 before Section 3.3 follows a straightforward logic. In Section 3.2, we introduce the PAC-Bayes theorem, which covers both the PAC-Bayes bound and the PAC-Bayes C-bound, laying the groundwork for Section 3.3. In Section 3.3, we discuss PAC-Bayes learning, which utilizes the PAC-Bayes bound as its objective function.
>
> **C4** In section 3.2, does the loss function need to be bounded in order for Theorem 3.1 to apply? As currently written, this isn't clear (though the binary KL divergence doesn't make sense if loss isn't in [0,1]).
>
> **R4** Thank you for bringing this to our attention. Indeed, a bounded loss function is required for Theorem 3.1. We have now made a note of this in the text.

---

### Review · Reviewer_4k6L · 2023-04-09

**Summary Of Contributions:**

sorry for the late review, but here it is.

The authors analyze a probabilistic training objective similar to (generalized) mean field variational inference objectives with gradient flow analysis, finding that there’s two separate neural tangent kernels (one for the mean parameters and one for the variance parameters) required to be considered for the analysis. However, the variance term washes out in the analysis of the mean function. Due to the training objective, the authors immediately get PAC-Bayesian generalization and risk bounds for the infinite neural networks that they analyze. They then are able to do some zero-shot hyperparameter tuning with these bounds.


**Audience:**

Yes

**Claims And Evidence:**

Yes

**Requested Changes:**

Major: What is the connection with the broader “infinite width Bayesian NNs are Gaussian Processes?” literature. It seems like the analysis in Novak et al, ’19 (“Bayesian Deep convolutional networks with many channels…”) is more “exact” in the sense that the limiting function is derived for any Bayesian neural network under these conditions, not specifically under your PAC Bayesian objective.

Perhaps more similarly, both Yang et al, and Immer et al (cited above) do zero-shot hyperparameter tuning with variants of the tangent kernel derived here. Specifically, Yang et al even study scaling laws, enabling the transfer of hyperparmeters and architectures from small models to much larger ones. Can such an analysis be done with your work?

Medium: What is the connection with compressibility bounds for generalization (“PAC Bayes Compression bounds so tight they can explain generalization,” Lotfi et al, ’22; “non vacuous generalization bounds at the imagenet scale…,” Zhou et al, ’19) in the PAC Bayesian framework? They seem to be quite useful now, but your analysis, from what I can tell, uses the standard PAC Bayesian bounds and doesn’t rely on compressibility of the network.

Minor: plot bottom row, right two figures on a log scale since the points are all bunch up. Maybe also plot the trend line and the dashed ‘1-1’ line.

minor: eq 2: is W_\sigma^{(l)} a vector or a matrix. Make this more clear please

minor: eq 10, why is setting the learning rate = 1 fine?

Minor: eq 16: I believe that \lambda is \lambda_0?

Minor: thm 4.3: doesn’t this just become standard Gaussian process PAC Bayesian generalization analysis at this point, as this is a standard GP now. I believe this is analogous to reference Nitanda and Suzuki, ’20.


**Strengths And Weaknesses:**

Strengths:

I like the swift move from risk bounds to zero shot hyperparameter tuning, which seems like a clear and compelling practical use case for these types of bounds.

The zero shot hyperparameter tuning experiments look very strong and convincing.

Weaknesses:

It feels like there’s a bunch of related work that’s missing and should be compared to (maybe this is due to the area being reasonably crowded):

Zero shot hyperparameter tuning from the NTK: “Tensor Programs V: …”, Yang et al, ‘22

Model selection from a Bayesian variant of the NTK: “Scalable Marginal Likelihood Estimation for Model Selection … “ Immer et al, ‘21

PAC Bayesian generalization bounds:

Dzugaite and Roy ’17 analyze and train a very similar objective to yours, except that they compute instead the average over the training set with random samples, instead of your sampled mean. This enables both training and blind hyperparameter tuning using their bound. Maybe the authors could compare with their approach?

My understanding is that Eq 7 comes very close to their training objectives, and to reparameterization VI training objectives as well. However, you train on loss(E(net(data)), while the VI training objectives are E(loss(net(data)) and Dzugaite and Roy train on a single sample.

---

> ### Author Response · Authors · 2023-04-18
> **Rebuttal by Authors Part I**
>
> Thank you for your time and your positive feedback! Please see our detailed responses to your comments and suggestions below.
>
> **W1** It feels like there’s a bunch of related work that’s missing and should be compared to (maybe this is due to the area being reasonably crowded): [1,2]
>
> **A1** Thank you for pointing out the related works. Both Yang et al. (2022) and Immer et al. (2021) focus on hyperparameter (HP) tuning and model selection. Specifically, Yang et al. (2022) initially parametrize the target model using Maximal Update Parametrization (µP). They then tune a smaller version (in terms of width and/or depth) of the target model and copy the tuned hyperparameters to the target model. Their approach relies on a scaling invariant property, while our method is based on a training invariant property. Immer et al. (2021) develop their method using Laplace's method and Gauss-Newton approximations to the Hessian. They update the differentiable hyperparameters of the model using gradients during training. After training, they use the marginal likelihood of the trained model to make discrete choices, such as selecting the network architecture or choosing between several trained models. In contrast to both methods, which involve training, our approach is training-free. We have included a comparison in the updated manuscript.
>
> **W2** PAC Bayesian generalization bounds: Dzugaite and Roy ’17 [3] analyze and train a very similar objective to yours, except that they compute instead the average over the training set with random samples, instead of your sampled mean. This enables both training and blind hyperparameter tuning using their bound. Maybe the authors could compare with their approach? My understanding is that Eq 7 comes very close to their training objectives, and to reparameterization VI training objectives as well. However, you train on loss(E(net(data)), while the VI training objectives are E(loss(net(data)) and Dzugaite and Roy train on a single sample.
>
> **A2** Your understanding is correct. We have modified the original target function to Eq. 7 in accordance with our theoretical framework. Specifically, we employ the empirical loss of the expected output function and set ℓ to be the squared loss in the training objective function. This approach offers the advantage of yielding an explicit solution for the expected output function in the infinite-width limit.
>
> **Major**: What is the connection with the broader “infinite width Bayesian NNs are Gaussian Processes?” literature. It seems like the analysis in Novak et al, ’19 (“Bayesian Deep convolutional networks with many channels…”) is more “exact” in the sense that the limiting function is derived for any Bayesian neural network under these conditions, not specifically under your PAC Bayesian objective. Perhaps more similarly, both Yang et al [1], and Immer et al [2] do zero-shot hyperparameter tuning with variants of the tangent kernel derived here. Specifically, Yang et al even study scaling laws, enabling the transfer of hyperparmeters and architectures from small models to much larger ones. Can such an analysis be done with your work?
>
> We would like to clarify that our method incorporates the result of "infinite width Bayesian NNs are Gaussian Processes" at initialization (without training). This result corresponds to the behavior of infinite width Bayesian NNs at initialization. However, our focus is on the training of "infinite width Bayesian NNs" using the PAC-Bayes bound, which is governed by the PNTK (our result).
>
> Generally, both Yang et al. and Immer et al. (cited above) can be applied to our case. However, we would like to emphasize that our method is training-free, which offers significant advantages in terms of time efficiency. While the other methods involve training processes that can be computationally expensive, our approach bypasses these time-consuming steps and allows for faster hyperparameter tuning or model selection. This distinction makes our method a more efficient option for certain scenarios where time cost is a major concern.

---

> > ### Comment · Reviewer_4k6L · 2023-04-23
> > **Clarification**
> >
> > >  We have modified the original target function to Eq. 7 in accordance with our theoretical framework.
> >
> > Could the authors explain this decision and derivation in a bit more detail?
> >
> > I think it's alluded to in the discussion with Q6fj but am not entirely sure.

---

> > > ### Author Response · Authors · 2023-04-25
> > > **Author Responses to Further Questions**
> > >
> > > We appreciate your additional comments and suggestions. Our responses are provided below:
> > >
> > > **Q1** Could the authors explain this decision and derivation in a bit more detail?
> > >
> > > **A1** Contrasting the generalization bound presented by [Dziugaite and Roy '17], which computes the average over the training set using random samples, our approach utilizes the empirical loss of the expected output function inspired by the PAC-Bayes $\mathcal{C}$-bound. In particular, the empirical loss $ \underline{R}\_\mathcal{S}(Q) \triangleq \frac{1}{n} \sum_{i=1}^{n} \ell \left( y_{i}, \hat{f} \left( {\bf x}{i} \right) \right) $ used in Equation (7) is a lower bound of the expected empirical loss (also known as Gibbs empirical risk) $R_\mathcal{S}(Q)$, as per Jensen's inequality.  Examining the risk of expected output within the PAC-Bayes
> > > bound presents the benefit of providing an explicit solution for optimization analysis in the infinite-width
> > > limit. Moreover, we can formulate the generalization bound of PAC-Bayesian learning upon convergence.
> > >
> > > We have updated the manuscript with revisions to the discussion and explanation, which are highlighted in purple.

---

> ### Author Response · Authors · 2023-04-18
> **Rebuttal by Authors Part II**
>
> **Medium**: What is the connection with compressibility bounds for generalization (“PAC Bayes Compression bounds so tight they can explain generalization,” Lotfi et al, ’22; “non vacuous generalization bounds at the imagenet scale…,” Zhou et al, ’19) in the PAC Bayesian framework? They seem to be quite useful now, but your analysis, from what I can tell, uses the standard PAC Bayesian bounds and doesn’t rely on compressibility of the network.
>
> Compression bounds are designed to scale the PAC-Bayes bound to larger datasets, such as ImageNet. These bounds aim to provide a more efficient and practical approach for dealing with massive amounts of data, which is commonly encountered in real-world applications. Our training-free method for hyperparameter tuning offers a promising alternative for such scenarios, as it can significantly reduce time cost without sacrificing performance. However, extending our approach to incorporate compression techniques and adapting it for use with large datasets like ImageNet is an interesting direction for future research. By exploring this avenue, we hope to make our method even more powerful and applicable to a broader range of tasks in deep learning.
>
> **Minor** : plot bottom row, right two figures on a log scale since the points are all bunch up. Maybe also plot the trend line and the dashed ‘1-1’ line.
>
> Thanks for your suggestion. We have incorporated the suggestion and created the plot. Please find it in the updated manuscript.
>
> **minor**: eq 2: is W_\sigma^{(l)} a vector or a matrix. Make this more clear please
>
> Thanks for pointing it out. $W_\sigma^{(l)}$ is a matrix and we have denoted it in the updated manuscript.
>
> **minor**: eq 10, why is setting the learning rate = 1 fine?
>
> For the gradient flow case, a learning rate of 1 is a typical setting.
>
> **Minor**: eq 16: I believe that \lambda is \lambda_0?
>
> It is $\lambda$, which represents the strength of KL regularization. In contrast, $\lambda_0$ is the smallest eigenvalue of a matrix.
>
> **Minor**: thm 4.3: doesn’t this just become standard Gaussian process PAC Bayesian generalization analysis at this point, as this is a standard GP now. I believe this is analogous to reference Nitanda and Suzuki, ’20.
>
> Thanks for pointing it out. We have cited and discussed our results in relation to Nitanda and Suzuki (2020) in our manuscript.
>
> **References**
>
> [1] Zero shot hyperparameter tuning from the NTK: “Tensor Programs V: …”, Yang et al, ‘22
>
> [2] Model selection from a Bayesian variant of the NTK: “Scalable Marginal Likelihood Estimation for Model Selection … “ Immer et al, ‘21
>
> [3] Dziugaite, Gintare Karolina, and Daniel M. Roy. "Computing nonvacuous generalization bounds for deep (stochastic) neural networks with many more parameters than training data." arXiv preprint arXiv:1703.11008 (2017). [4]

---

### Decision · Action_Editors · 2023-05-01

**Recommendation:** Accept with minor revision

**Comment:**

Dear authors,
After reading the reviews, (detailed -- thanks!) rebuttals and discussion, I am happy to recommend acceptance of your submission. I believe you make a significant contribution to theoretical analysis of deep learning through PAC-Bayes -- thank you for revising your work along the reviewers' suggestions and congratulations on a fine piece of work.
I am happy with most of the manuscript but I have however a request before the article gets published: the bibliography is a bit sloppy and requires some careful editing. Please capitalise properly all words that require it ('PAC' instead of 'pac', etc.) and do make sure your citations are accurate: I came across a few papers listed as arXiv preprint when they are in fact long published.
Best,
Benjamin

**Audience:**

I am convinced the paper will be of interest to a solid part of the machine learning community and TMLR's audience.

**Claims And Evidence:**

All claims (theorems, experimental findings) are supported by convincing evidence.

---

> ### Author Response · Authors · 2023-05-17
> **Response to decision and suggestions by the action editor**
>
> We would like to express our gratitude to the action editor for your insightful suggestions and recommendations.
>
> We've updated the manuscript to adhere to the proper capitalization rules and have replaced the preprint version previously available on Arxiv with the published version. For your convenience and further review, the final, camera-ready version has been uploaded.